# BEYOND DAGS: A LATENT PARTIAL CAUSAL MODEL FOR MULTIMODAL LEARNING

**Yuhang Liu[1,2], Zhen Zhang[1,2], Dong Gong[3], Erdun Gao[1,2], Biwei Huang[4],**
**Mingming Gong[5], Anton van den Hengel[1,2], Kun Zhang[6], Javen Qinfeng Shi[1,2]**
[1]Responsible AI Research Centre, Australia
[2]Australian Institute for Machine Learning, Adelaide University
[3]School of Computer Science and Engineering, The University of New South Wales
[4]Halıcıoğlu Data Science Institute, University of California San Diego
[5]School of Mathematics and Statistics, The University of Melbourne
[6]Department of Philosophy, Carnegie Mellon University
yuhang.liu01@adelaide.edu.au

**Project:** https://sites.google.com/view/yuhangliu/projects/bedags

## ABSTRACT

Directed Acyclic Graphs (DAGs) are a standard tool in causal modeling, but their suitability for capturing the complexity of large-scale multimodal data is questionable. In practice, real-world multimodal datasets are often collected from heterogeneous generative processes that do not conform to a single DAG. Instead, they may involve multiple, and even opposing, DAG structures with inverse causal directions. To address this gap, in this work, we first propose a novel latent partial causal model tailored for multimodal data representation learning, featuring two latent coupled variables parts connected by an undirected edge, to represent the transfer of knowledge across modalities. Under specific statistical assumptions, we establish an identifiability result, demonstrating that representations learned by MultiModal Contrastive Learning (MMCL) correspond to the latent coupled variables up to a trivial transformation. This result deepens our understanding of the why MMCL works, highlights its potential for representation disentanglement, and expands the utility of pre-trained models like CLIP. Synthetic experiments confirm the robustness of our findings, even when the assumptions are partially violated. Most importantly, experiments on a pre-trained CLIP model embodies disentangled representations, enabling few-shot learning and improving domain generalization across diverse real-world datasets. Together, these contributions push the boundaries of MMCL, both in theory and in practical applications.

## 1 INTRODUCTION

Recent advances in multimodal learning have demonstrated remarkable capabilities across vision, language, and beyond (Liang et al., 2024; Lymperaiou & Stamou, 2024; Li et al., 2024). Representative models, such as CLIP, achieve this by aligning different modalities through MultiModal Contrastive Learning (MMCL) (Radford et al., 2021). A crucial factor behind their success is that these models are trained on large-scale multimodal datasets, enabling them to learn rich, high-quality cross-modal representations. Despite its remarkable empirical success, understanding the underlying mechanisms of multimodal learning is essential, not only to explain its current achievements but also to identify opportunities for further improvements (Liang et al., 2024). Recent works have also analyzed multimodal learning through the lens of latent causal models (Daunhawer et al., 2023; Yao et al., 2024; Gresele et al., 2020). These approaches examine the relationship between representations learned by multimodal learning from observed data and the high-level latent causal variables underlying such data, a line of inquiry referred to as identifiability analysis. By demonstrating that learned representations can, in principle, recover these latent causal variables, such analyses provide a causality-grounded explanation for the success of multimodal models. *Crucially, most of these*

*latent causal models rely on the assumption that the latent causal variables follow a Directed Acyclic Graph (DAG) structure.* See Appendix A for more related work.

We argue that such a DAG assumption may be inappropriate for capturing the underlying generative processes of large-scale multimodal data, which underpin state-of-the-art multimodal models. This argument is supported by the following observation that large-scale multimodal data often arise from heterogeneous causal mechanisms that correspond to different, and sometimes even conflicting, DAG structures (Schölkopf et al., 2012). For instance, in the context of text–image paired data, some pairs are generated through a text-to-image causal mechanism, where a textual instruction serves as the input from which the corresponding image is produced (Ramesh et al., 2021). In contrast, some pairs arise from an image-to-text pipeline, where images are first collected from the internet and subsequently annotated with descriptive text by experts (Sharma et al., 2018). These two distinct causal mechanisms illustrate that large-scale multimodal data may arise from fundamentally opposite causal directions. Consequently, the common DAG assumption may be overly restrictive, failing to capture the diverse and sometimes conflicting generative processes underlying such data (see Sec. 2 for a detailed discussion). As a result, although prior works on identifiability analysis under DAG assumptions (Daunhawer et al., 2023; Yao et al., 2023; Gresele et al., 2020) have provided valuable theoretical insights, they are often restricted to specific, small-scale multimodal data, where a DAG structure is sufficient to capture the underlying generative process. As a direct consequence of this modeling choice, these studies largely remain confined to simulation experiments, and offer limited guidance for applying advanced multimodal models trained on large-scale data, e.g., CLIP-like models, to real-world applications. To this end, this paper makes the following contributions:

• *A Novel Latent Partial Causal Model (Sec. 2).* We propose a novel latent partial causal generative model, specifically designed for modeling the multimodal data generation process. Instead of relying on the DAGs assumption, our model introduces latent coupled variables, connected by undirected edges, to effectively capture transferable knowledge across different modalities.

• *Identifiability Guarantee (Sec. 3 and 4).* We developed theoretical analyses specifically tailored to the proposed generative model, under certain statistical assumptions, showing that the representations learned by MMCL are related to the latent coupled variables up to a simple transformation, thereby providing a theoretical explanation for the success of MMCL.

• *Disentanglement Potential of MMCL (Sec. 5).* Our theoretical results reveal the component-wise disentanglement potential of MMCL, which pushes the boundaries of how pre-trained models, such as CLIP-like models, can be leveraged. *To the best of our knowledge, this is the first work to provide guarantees for the component-wise disentanglement potential of MMCL.*

• *Extensive Experimental Results (Sec. 6).* We validate our theoretical findings under ideal conditions via simulations and demonstrate their robustness even when the underlying assumptions are partially violated. Extensive experiments on pre-trained CLIP model across various tasks, such as few-shot learning, domain generalization, and disentangled representation learning, on over 16 real-world datasets substantiate the practical effectiveness of our findings.

In summary, our work provides a principled explanation for the success of MMCL and, importantly, highlights its potential for learning disentangled representations. Although our theoretical findings rely on certain assumptions that may not be fully verifiable in practice, similar to most existing works on identifiability analysis, simulations demonstrate the robustness of our results even when these assumptions are partially violated. In addition, extensive experiments with pre-trained CLIP models across diverse real-world tasks provide strong evidence that the theoretical insights can translate into practical benefits. Taken together, these findings relax the conventional reliance on DAG assumptions in advanced MMCL, while maintaining applicability and effectiveness in real-world scenarios.

## 2    GENERATIVE MODEL: THE LATENT PARTIAL CAUSAL MODEL

In this section, we introduce a latent partial causal model that captures the generative mechanisms of multimodal data. Before presenting the model, we outline a key observation about such data.

**Diversity in generative process of large-scale multimodal data.**    We argue that real-world large-scale multimodal data often entails multiple, complex generative processes that may not be fully captured by a single DAG structure. To illustrate this (see Figure 1), let latent variables $\mathbf{z}_x$ and $\mathbf{z}_t$

Figure 1: Possible DAG structures underlying large-scale multimodal data: Left: A latent confounder influences both $\mathbf{z}_x$ and $\mathbf{z}_t$. Middle: $\mathbf{z}_t$ influences $\mathbf{z}_x$ through an intermediate mediator $\mathbf{b}$, serving as a bottleneck for transferable knowledge. Right: A symmetric inverse relationship where $\mathbf{z}_x$ influences $\mathbf{z}_t$ via $\mathbf{b}$. These DAGs illustrate that a single DAG assumption may not hold when modeling large-scale multimodal data with heterogeneous generative processes.

denote shared semantic factors. For example, $\mathbf{z}_x$ may correspond to high-level visual concepts such as object category or scene type in an image, while $\mathbf{z}_t$ may capture the semantic content of a sentence, such as topic or intent. To model modality-specific characteristics, we introduce additional latent variables $\mathbf{m}_x$ and $\mathbf{m}_t$. For instance, $\mathbf{m}_x$ may represent image-specific factors such as background noise or visual artifacts, whereas $\mathbf{m}_t$ could encode linguistic aspects such as sentence structure or grammatical patterns. Together, $(\mathbf{z}_x, \mathbf{m}_x)$ and $(\mathbf{z}_t, \mathbf{m}_t)$ generate the observed variables $\mathbf{x}$ (image) and $\mathbf{t}$ (text), respectively.

In the left DAG model of Figure 1, the latent confounder $\mathbf{c}$ represents a shared source of variation that influences both latent variables $\mathbf{z}_x$ and $\mathbf{z}_t$, which correspond to latent semantic factors generating the observed variables $\mathbf{x}$ (e.g., image) and $\mathbf{t}$ (e.g., text), respectively. This confounder captures a common underlying context or concept connecting the two modalities. For example, if the image and text are related to the topic "sports," $\mathbf{c}$ could encapsulate this shared theme, influencing the generation of both the visual and textual data. The middle DAG depicts a structure where $\mathbf{b}$ represents transferable knowledge. Specifically, $\mathbf{b}$ serves as the bridge, deriving information from the text latent variable $\mathbf{z}_t$ and informing the image latent space $\mathbf{z}_x$. This scenario aligns with the generative process where text serves as a guiding input for image generation, e.g., text-to-image generation. A classical example is the MNIST dataset (LeCun & Cortes, 2005). In contrast, the DAG on the right represents an image-guided text generation process, e.g., image captioning. Here, the high-level latent information in the image influences the high-level latent variable in the generated caption. A classical example is the CelebA dataset (Jiang et al., 2021).

Current advanced multimodal models, such as CLIP (Liang et al., 2022), are typically trained on vast collections of multimodal data, which may in fact arise from a mixture of the three scenarios illustrated in Figure 1 (potentially with additional DAG assumptions not depicted). In this context, restricting the generative modeling of large-scale multimodal data to a single DAG structure may be inadequate to capture the inherent diversity of real-world multimodal dependencies.

**The Proposed Latent Partial Causal Models.** Instead of DAGs structure, we propose latent partial causal model, designed to represent the generative process for multimodal data, as illustrated in Figure 2. In it, the latent space is partitioned into two components, each corresponding to a specific modality, such as image and text. To capture unique characteristics within each domain, the model incorporates modality-specific latent variables, $\mathbf{m}_x$ and $\mathbf{m}_t$. In addition, to capture transferable knowledge between these modalities, the model introduces an undirected edge between the latent coupled variables, $\mathbf{z}_x$ and $\mathbf{z}_t$. Further, the observations are generated through distinct processes that link the latent variables to the observed data. Specifically, images ($\mathbf{x}$) are generated by the function

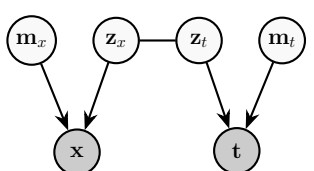

Figure 2: The proposed latent partial causal model. $\mathbf{z}_x$ and $\mathbf{z}_t$ are latent coupled variables, and $\mathbf{m}_x$, $\mathbf{m}_t$ are modality-specific.

$\mathbf{g}_x(\mathbf{m}_x, \mathbf{z}_x)$, while text ($\mathbf{t}$) is produced by $\mathbf{g}_t(\mathbf{m}_t, \mathbf{z}_t)$. Besides the justification mentioned in Figure 1, this modeling approach is also grounded in the intricate dependencies between modalities. For instance, the adage "a picture is worth a thousand words" highlights the richness and detail of visual data, as supported by Gropper (1963); Hum et al. (2011). However, this perspective is not universally applicable, as Reinert (1976) argues that textual information can often convey more precise meanings. Similarly, Fidler et al. (2013) reinforces the complementary nature of text, asserting that "a sentence is worth a thousand pixels" in its ability to succinctly express complex ideas.

## 3   A FIRST LOOK: THE RECOVERY POTENTIAL OF MMCL

Given the proposed generative model, our goal is to analyze how MMCL framework, trained with observed data $\mathbf{x}$ and $\mathbf{t}$, can recover the true latent variables $\mathbf{z}_x$ and $\mathbf{z}_t$, up to a simple transformation. Before this, we provide an intuitive motivation for why MCL is expected to achieve this. MMCL leverages a loss function designed to maximize similarity between embeddings of real paired data while minimizing similarity for incorrect pairs. The loss function is defined as Zhang et al. (2022b); Radford et al. (2021):

$$\mathcal{L} = -\frac{1}{N}\sum_{i=1}^{N}\log\frac{e^{-d\left(\mathbf{f}_x(\mathbf{x}_i),\mathbf{f}_t(\mathbf{t}_i)\right)/\tau}}{\sum_{j=1}^{N}e^{-d\left(\mathbf{f}_x(\mathbf{x}_i),\mathbf{f}_t(\mathbf{t}_j)\right)/\tau}} - \frac{1}{N}\sum_{i=1}^{N}\log\frac{e^{-d\left(\mathbf{f}_x(\mathbf{x}_i),\mathbf{f}_t(\mathbf{t}_i)\right)/\tau}}{\sum_{j=1}^{N}e^{-d\left(\mathbf{f}_x(\mathbf{x}_j),\mathbf{f}_t(\mathbf{t}_i)\right)/\tau}}, \tag{1}$$

where $d$ denotes a distance metric, *e.g.*, cosine similarity on hypersphere or L1 norm on convex bodies, $\tau$ is a learnable temperature hyper-parameter, $N$ denotes the sample size, which means that we have $N$ positive pairs and $N^2 - N$ negative pairs, $\mathbf{f}_x$ denote the encoder on one modality $\mathbf{x}$, *i.e.*, image, similarly, $\mathbf{f}_t$ denote the encoder on another $\mathbf{t}$, *i.e.*, text. To understand the multimodal contrastive loss further, we investigate its asymptotics:

**Theorem 3.1** (Asymptotics of $\mathcal{L}$). *For fixed $\tau > 0$, as the sample size $N \to \infty$, the (normalized) multimodal contrastive loss converges to*

$$\lim_{N\to\infty}\mathcal{L} - 2\log N = 2\mathop{\mathbb{E}}_{(\mathbf{x},\mathbf{t})\sim p(\mathbf{x},\mathbf{t})}\left[d\left(\mathbf{f}_x(\mathbf{x}),\mathbf{f}_t(\mathbf{t})\right)/\tau\right] + \mathop{\mathbb{E}}_{\mathbf{x}\sim p(\mathbf{x})}\left[\log\mathop{\mathbb{E}}_{\mathbf{t}\sim p(\mathbf{t})}\left[e^{-d\left(\mathbf{f}_x(\mathbf{x}),\mathbf{f}_t(\mathbf{t})\right)/\tau}\right]\right]$$
$$+ \mathop{\mathbb{E}}_{\mathbf{t}\sim p(\mathbf{t})}\left[\log\mathop{\mathbb{E}}_{\mathbf{x}\sim p(\mathbf{x})}\left[e^{-d\left(\mathbf{f}_x(\mathbf{x}),\mathbf{f}_t(\mathbf{t})\right)/\tau}\right]\right]. \tag{2}$$

The proof is provided in Appendix B. This is a generalization of Theorem 1 in Wang & Isola (2020).

**Insights into Latent Variable Recovery**   The loss function in Eq. (2) connects directly to two fundamental principles in latent variable recovery: Prior Matching and Information Preservation. These principles are crucial for methods like nonlinear independent component analysis (ICA) (Hyvärinen et al., 2001), which recover latent independent variables from observed data.

- *Prior Matching*: This constrains the solution space using prior knowledge, addressing the non-uniqueness problem that often arises in latent variable recovery.

- *Information Preservation*: This ensures that the solution space fully captures the complexity of the latent variables derived from the observed data.

**Prior Matching**   The first term in Eq. (2) promotes alignment between representations of real data pairs across modalities, enforcing that one modality (i.e., text) acts as a prior signal for the other (i.e., image). Minimizing this term drives cross-modal alignment and incorporates prior knowledge, which is key for recovering latent variables.

**Information Preservation**   The last two terms in Eq. (2) are closely related to ensuring that the learned representations capture the full complexity of the latent variables. These terms can be approximated by optimizing the following expression (proof in Appendix C):

$$-H\left(p(\mathbf{f}_x(\mathbf{x})),p(\mathbf{f}_t(\mathbf{t}))\right) - H\left(p(\mathbf{f}_t(\mathbf{t})),p(\mathbf{f}_x(\mathbf{x}))\right), \tag{3}$$

where $H(\cdot,\cdot)$ denotes cross-entropy. The objective function in Eq. (2) is symmetric between $\mathbf{x}$ and $\mathbf{t}$. Intuitively, if $p(\mathbf{f}_x(\mathbf{x}))$ and $p(\mathbf{f}_t(\mathbf{t}))$ are not equal, the solution deviates, increasing the objective value and introducing asymmetry in the last two terms. For the optimal solution, the two distributions must align. When $p(\mathbf{f}_x(\mathbf{x})) = p(\mathbf{f}_t(\mathbf{t}))$, the cross-entropy in Eq. (3) reduces to entropy, and if $\mathbf{f}_x$ and $\mathbf{f}_t$ transform $\mathbf{x}$ and $\mathbf{t}$ into uniformly distributed random variables, Eq. (3) reaches its optimal value. This highlights the importance of finding transformations $\mathbf{f}_x$ and $\mathbf{f}_t$ that preserve information by fully capturing the latent variable structure.

**A Novel Unified Perspective on Contrastive Loss** Previous research has primarily focused on contrastive loss in the context of single modality, emphasizing two main perspectives: 1) alignment-uniformity (Wang & Isola, 2020), which is closely related to prior matching, and 2) information preservation (Oord et al., 2018). However, these two perspectives have largely been treated separately. In this work, we offer a novel insight by combining these two perspectives within the multimodal context for latent variable recovery. This insight motivates our belief that MMCL holds significant potential for recovering latent variables.

## 4 FROM POTENTIAL TO PRINCIPLES: IDENTIFIABILITY GUARANTEE

Given the initiative analysis in Section 3, which highlights the potential of MMCL for recovering latent variables, we now move forward to rigorous identifiability analysis, which provide theoretical guarantees that MMCL can indeed recover the true latent variables, by parameterizing the proposed latent partial causal model. We examine two distinct types of parameterization in latent spaces, hyperspheres and convex bodies, under specific assumptions, respectively.

### 4.1 IDENTIFIABILITY ANALYSIS ON HYPERSPHERE

On hypersphere, we parameterize the proposed latent partial causal generative models as following:

$$p(\mathbf{z}_x) = |\mathcal{Z}|^{-1}, \quad p(\mathbf{z}_t|\mathbf{z}_x) = C_p^{-1} e^{(k\mathbf{z}_t^T \mathbf{z}_x)}, \quad \mathbf{x} = \mathbf{g}_x(\mathbf{z}_x, \mathbf{m}_x), \quad \mathbf{t} = \mathbf{g}_t(\mathbf{z}_t, \mathbf{m}_t), \tag{4}$$

where $\mathcal{Z}$ denotes the space of latent factors $\mathbf{z}_x$ and $\mathbf{z}_t$. We assume that $\mathcal{Z}$ is the unit hypersphere $\mathbb{S}^{M-1}$, aligning with the commonly used normalization in constrastive loss. We do not enforce any further assumptions for $\mathbf{m}_x$ and $\mathbf{m}_t$. For $\mathbf{g}_x$ and $\mathbf{g}_t$, we assume them to be invertible, and differentiable, ensuring the information in latent space can be recovered. In addition, we assume that $p(\mathbf{z}_x)$ follows a uniform distribution, and $p(\mathbf{z}_t|\mathbf{z}_x)$ follows a von Mises-Fisher (vMF) distribution, considering the constraint of unit hypersphere. Given these assumptions, we first establish that the minimization of the cross-entropy Eq. (2) converges to a symmetric cross entropy, as follows:

**Theorem 4.1.** ($\mathcal{L}$ *converges to the symmetric cross-entropy*) *Under the assumptions defined in Eqs.* (4) *for the proposed latent partial causal model, the necessary condition* $\mathbf{f}_x \circ \mathbf{g}_x = \mathbf{f}_t \circ \mathbf{g}_t$, *denoted as* $\mathbf{h}$, *for the optimal normalized multimodal contrastiveloss given by Eq.* (2) *leads to the following reduction of the loss itself:*

$$\lim_{N \to \infty} \mathcal{L} - 2\log N + 2\log|\mathcal{Z}| = \mathbb{E}_{\mathbf{z}_x \sim p(\mathbf{z}_x)} \Big[ H(p(\mathbf{z}_t|\mathbf{z}_x), q_{\mathbf{h}}(\mathbf{z}_t|\mathbf{z}_x)) \Big] + \mathbb{E}_{\mathbf{z}_t \sim p(\mathbf{z}_t)} \Big[ H(p(\mathbf{z}_x|\mathbf{z}_t), q_{\mathbf{h}}(\mathbf{z}_x|\mathbf{z}_t)) \Big],$$

$$\tag{5}$$

*where* $H$ *is the cross entropy, the conditional distributions* $q_{\mathbf{h}}(\mathbf{z}_t|\mathbf{z}_x)$ *and* $q(\mathbf{z}_x|\mathbf{z}_t)$ *are parameterized by the following:*

$$q_{\mathbf{h}}(\mathbf{z}_x|\mathbf{z}_t) = C_q(\mathbf{z}_t)^{-1} e^{(\mathbf{h}(\mathbf{z}_x)^T \mathbf{h}(\mathbf{z}_t)/\tau)} q_{\mathbf{h}}(\mathbf{z}_t|\mathbf{z}_x) = C_q(\mathbf{z}_x)^{-1} e^{(\mathbf{h}(\mathbf{z}_t)^T \mathbf{h}(\mathbf{z}_x)/\tau)}, \tag{6}$$

*with*

$$C_q(\mathbf{z}_t) = \int e^{(\mathbf{h}(\mathbf{z}_x)^T \mathbf{h}(\mathbf{z}_t)/\tau)} \mathrm{d}\mathbf{z}_x, C_q(\mathbf{z}_x) = \int e^{(\mathbf{h}(\mathbf{z}_x)^T \mathbf{h}(\mathbf{z}_t)/\tau)} \mathrm{d}\mathbf{z}_t.$$

Refer to Appendix D.1 for proof. This is a generalization of Theorem 1 in Zimmermann et al. (2021).

**Bridge Between Modalities** By addressing key asymmetries arising from modality differences, such as modality-specific variables $\mathbf{m}_x$ and $\mathbf{m}_t$, along with distinct generative processes $\mathbf{g}_x$ and $\mathbf{g}_t$, we derive the result in Theorem 4.1. This result is pivotal as it establishes a critical connection between MMCL and traditional single-modal contrastive learning. In particular, Theorem 4.1 enables the transfer of insights and results from single-modal settings to the multimodal context. As a result, we present the following corollary:

**Corollary 1.** *By leveraging Theorem 4.1, the minimization of Eq.* (5) *identifies the latent variables* $\mathbf{z}_x$ *(and symmetrically,* $\mathbf{z}_t$*) up to a linear transformation. Specifically, the representations* $\mathbf{f}_x(\mathbf{x})$, *learned by the minimization of Eq.* (5), *are linearly related to the underlying latent variables* $\mathbf{z}_x$ *in the proposed latent partial causal model, as follows:* $\mathbf{f}_x(\mathbf{x}) = \mathbf{A}\mathbf{z}_x + \mathbf{c}$, *where* $\mathbf{A}$ *is an orthogonal matrix and* $\mathbf{c}$ *is a constant vector.*

For further details, see Appendix D.2.

**Success of MMCL**    Corollary 1 shows that minimizing Eq. (5) (or equivalently, the multimodal contrastive loss in Eq. (1)) identifies the latent variables $\mathbf{z}_x$ (and symmetrically, $\mathbf{z}_t$) up to a linear transformation. This means that the representations $\mathbf{f}_x(\mathbf{x})$, learned through MMCL, are directly related to the latent variables $\mathbf{z}_x$ via a linear transformation, i.e., $\mathbf{f}_x(\mathbf{x}) = \mathbf{A}\mathbf{z}_x + \mathbf{c}$. A similar result holds for $\mathbf{z}_t$. This finding highlights the effectiveness of MMCL, suggesting that its success in practical applications stems from its ability to recover latent coupled variables. This recovery preserves essential, transferable knowledge across modalities, enabling the learned representations to capture high-level transferable information while discarding model-specific details. Such properties are key to the robustness and transferability of MMCL representations.

## 4.2 IDENTIFIABILITY ANALYSIS ON CONVEX BODIES

We now extend the previous identifiability result to convex bodies, *e.g.*, the hyperrectangle $[a_1, b_1] \times \ldots \times [a_M, b_M]$. On convex bodies, we parameterize the proposed generative models by the following:

$$p(\mathbf{z}_x) = |\mathcal{Z}_c|^{-1}, \quad p(\mathbf{z}_t|\mathbf{z}_x) = C_p(\mathbf{z}_x)^{-1} e^{-\delta(\mathbf{z}_t, \mathbf{z}_x)/\lambda}, \quad \mathbf{x} = \mathbf{g}_x(\mathbf{z}_x, \mathbf{m}_x), \quad \mathbf{t} = \mathbf{g}_t(\mathbf{z}_t, \mathbf{m}_t), \quad (7)$$

where $\delta$ is a distance metric induced by a norm. We consider a convex body in $\mathbb{R}^M$, denoted as $\mathcal{Z}_c$, where we assume that $p(\mathbf{z}_x)$ follows a uniform distribution, and the conditional distribution $p(\mathbf{z}_t|\mathbf{z}_x)$ follows an exponential distribution. Again, we do not enforce any further assumptions for $\mathbf{m}_x$ and $\mathbf{m}_t$. For $\mathbf{g}_x$ and $\mathbf{g}_t$, we assume them to be invertible and differentiable mapping, ensuring information in latent space can be recovered. Given these assumptions, we have the following result:

**Theorem 4.2.** *($\mathcal{L}$ converges to the symmetric cross-entropy) Under the assumptions defined in Eq.* (7) *for the proposed latent partial causal model, the necessary condition $\mathbf{f}_x \circ \mathbf{g}_x = \mathbf{f}_t \circ \mathbf{g}_t$, denoted as $\mathbf{h}$, for the optimal normalized multimodal contrastiveloss given by Eq.* (2) *leads to the following reduction of the loss itself:*

$$\lim_{N \to \infty} \mathcal{L} - 2\log N + 2\log|\mathcal{Z}_c| = \mathop{\mathbb{E}}_{\mathbf{z}_x \sim p(\mathbf{z}_x)} \left[ H(p(\mathbf{z}_t|\mathbf{z}_x), q_{\mathbf{h}}(\mathbf{z}_t|\mathbf{z}_x)) \right] + \mathop{\mathbb{E}}_{\mathbf{z}_t \sim p(\mathbf{z}_t)} \left[ H(p(\mathbf{z}_x|\mathbf{z}_t), q_{\mathbf{h}}(\mathbf{z}_x|\mathbf{z}_t)) \right],$$
$$(8)$$

*where $H$ is the cross entropy, the conditional distributions $q_{\mathbf{h}}(\mathbf{z}_t|\mathbf{z}_x)$ and $q(\mathbf{z}_x|\mathbf{z}_t)$ are parameterized by the following:*

$$q_{\mathbf{h}}(\mathbf{z}_x|\mathbf{z}_t) = C_q(\mathbf{z}_t) e^{-\delta(\mathbf{h}(\mathbf{z}_x), \mathbf{h}(\mathbf{z}_t))/\tau}, q_{\mathbf{h}}(\mathbf{z}_t|\mathbf{z}_x) = C_q(\mathbf{z}_x) e^{-\delta(\mathbf{h}(\mathbf{z}_x), \mathbf{h}(\mathbf{z}_t))/\tau}, \quad (9)$$

*with*

$$C_q(\mathbf{z}_t) = \int e^{-\delta(\mathbf{h}(\mathbf{z}_x), \mathbf{h}(\mathbf{z}_t))/\tau} \mathrm{d}\mathbf{z}_x, C_q(\mathbf{z}_x) = \int e^{-\delta(\mathbf{h}(\mathbf{z}_x), \mathbf{h}(\mathbf{z}_t))/\tau} \mathrm{d}\mathbf{z}_t.$$

**Bridge Between Modalities**    In convex bodies, Theorem 4.2, introduced for the first time in this work, plays a key role in bridging MMCL with traditional contrastive learning by addressing the asymmetric challenges arising from modality differences. Building on this theorem, we have:

**Corollary 2.** *The minimization of Eq.* (8) *in theorem 4.2 identifies the latent variables $\mathbf{z}_x$ (symmetrically, $\mathbf{z}_t$) up to a permutation transformation, i.e., the representations $\mathbf{f}_x(\mathbf{x})$, learned by the minimization of Eq.* (8)*, is related to the underlying $\mathbf{z}_x$ in the proposed partial causal model as follows: $\mathbf{f}_x(\mathbf{x}) = \mathbf{P}\mathbf{z}_x + \mathbf{c}$, where $\mathbf{P}$ is an permutation matrix with scaling, $\mathbf{c}$ is a constant vector.*

For completeness, see details in Appendix E.2.

**Success of MMCL**    Similar to Corollary 1 on hyperspheres, Corollary 2 establishes that, on convex bodies, the representations $\mathbf{f}_x(\mathbf{x})$ learned by MMCL are related to the true latent variables $\mathbf{z}_x$ as $\mathbf{f}_x(\mathbf{x}) = \mathbf{P}\mathbf{z}_x + \mathbf{c}$. This provides a foundation for the success of MMCL on convex bodies.

## 5 FROM PRINCIPLES TO PRACTICE: DISENTANGLEMENT IN CLIP MODELS

In theory, both Corollaries 1 and 2 suggest a disentanglement potential of CLIP-like models trained by MMCL, under the assumption that the variables in $\mathbf{z}_x$ (and symmetrically, $\mathbf{z}_t$) are mutually independent. we explore how these theoretical insights can be translated into practical guidance for the effective use of CLIP-like models.

Corollary 1 shows that the representations $\mathbf{f}_x(\mathbf{x})$ learned by MMCL are linearly related to the true latent variables $\mathbf{z}_x$ via an orthogonal transformation, i.e., $\mathbf{f}_x(\mathbf{x}) = \mathbf{A}\mathbf{z}_x + \mathbf{c}$. This result holds under two key conditions: (1) the true latent variables are sampled from a hyperspherical latent space, and (2) the inference model, e.g., CLIP-like models, is trained in a hyperspherical inference space. Notably, CLIP-like models naturally satisfy condition (2), as they typically employ L2 normalization, constraining representations to the unit sphere. Therefore, under condition (1) holds, the representations from CLIP-like models can be passed through linear unmixing method (e.g., FastICA (Hyvarinen, 1999)) to resolve the mixing matrix $\mathbf{A}$, resulting in disentangled representations. It is worth to note that the geometry of the hypersphere, specifically the unit $M-1$-dimensional hypersphere, places an upper bound on the number of independent variables, i.e., $M-1$ at most.

Unlike Corollary 1, Corollary 2 shows that the learned representations $\mathbf{f}_x(\mathbf{x})$ from MMCL are already disentangled, i.e., $\mathbf{f}_x(\mathbf{x}) = \mathbf{P}\mathbf{z}_x + \mathbf{c}$. This result mainly requires two conditions: (1) the true latent variables $\mathbf{z}_x$ are sampled from a convex body latent space, and (2) the inference space of the model is also constrained to a convex body. However, CLIP-like models typically violate the second condition, as they operate in a hyperspherical inference space due to L2 normalization, even through we assume that the first condition may hold. Nevertheless, the insight from Corollary 2 remains useful with appropriate adjustments. In particular, the Corollary relies on the existence of an isometric mapping from the latent space to the representation space (see Eq. (48) in Appendix). Although a global isometry from a convex body to a hypersphere is not feasible, it is reasonable to assume a local isometry between the convex body and small regions of the hypersphere. Based on this, we propose first applying Principal Component Analysis (PCA) to the representations $\mathbf{f}_x(\mathbf{x})$. Then, FastICA can be used to account for the orthogonal transformation introduced by PCA, enabling the extraction of the final disentangled representations. This PCA+ICA pipeline thus enables effective use of CLIP-like models under the result of Corollary 2.

**Remark 1.** *By leveraging the disentanglement capabilities of CLIP-like models, we can improve performance on tasks that benefit from disentangled representations, such as few-shot learning and domain generalization. This observation further motivates exploration of the disentanglement potential inherent in CLIP-like models across a broad range of downstream applications.*

## 6    SYNTHETIC EXPERIMENTS AND REAL-WORLD EVALUATION

**Synthetic Experiments**    In our initial experiments, we use synthetic data to validate our main identifiability results on hyperspheres and convex bodies, while also empirically assessing their robustness under significant violations of assumptions. We first sample $p(\mathbf{z}_x)$ according to the distributions listed in Table 1. Additionally, we generate paired samples from the conditional distribution $p(\mathbf{z}_t|\mathbf{z}_x)$ following the distributions specified in the same table. Beyond hyperspheres, our experiments also consider bounded and unbounded spaces. Each experiment is repeated three times for every setting. For more details regarding experiments, refer to Appendix K.

To evaluate linear identifiability result in Corollary 1, we fit a linear regression model between the ground-truth $\mathbf{z}_x$ and representations $\mathbf{f}_x(\mathbf{x})$ learned by MMCL and report the coefficient of determination ($R^2$). Further, to evaluate permutation identifiability result in Corollary 2, we employ the mean correlation coefficient (MCC) between the ground-truth $\mathbf{z}_x$ and representations $\mathbf{f}_x(\mathbf{x})$ learned by MMCL. The first row in Table 1 (left) and the first two rows in Table 1 (right), corresponding to the setting where the assumptions are satisfied, verify the identifiability results on hypersphere and convex bodies, respectively. Our empirical investigations have yielded a critical insight: discrepancies in the assumptions concerning marginal and conditional distributions, as well as the nature of the spaces (hypersphere and convex body), do not significantly impact performance. This robustness is demonstrated by the results detailed in Table 1 (left) for the hypersphere space and Table 1 (right) for convex bodies. This might be attributed to the fact that the loss function described in Eq. (2) predominantly relies on expectation computations, inherently allowing for a wide range of approximations. If we can approximate the expectation calculations consistently across various distributions and spaces, it is reasonable that the identifiability results remain well within acceptable bounds.

**Real-World Evaluation with Pretrained CLIP**    In real data, the true latent coupled variables are unknown. Therefore, we evaluate our theoretical findings from the perspective of disentanglement as discussed in Section 5. *Again, we emphasize that, in contrast to previous studies on identifiability for MMCL (Daunhawer et al., 2023; Yao et al., 2023; Gresele et al., 2020), which rely on simulation*

Table 1: Assessing identifiability up to linear (left) and permutation (right) transformations under varying assumptions. The first row in (left) and the first two rows in (right) represent settings that align with our assumptions in Corollary 1 and Corollary 2, while the others show results for violated assumptions. S: Space, Sp: Sphere, U: Uniform, v: vMF ($k = 1$), L: Laplace ($\lambda = 0.05$), N: Normal ($\delta = 0.05$), B: Box, Un: Unbounded, G: GenNorm ($\beta = 3$).

| Generative process | | | Model | | |
|---|---|---|---|---|---|
| S | $p(\mathbf{z}_x)$ | $p(\mathbf{z}_x|\mathbf{z}_t)$ | S | $q(\mathbf{z}_x|\mathbf{z}_t)$ | $R^2$ |
| Sp | U | v | Sp | v | 99.5 ± 0.1 |
| Sp | U | L | Sp | v | 99.4 ± 0.2 |
| Sp | U | N | Sp | v | 98.7 ± 0.3 |
| B | U | N | Un | N | 90.5 ± 0.2 |
| B | U | L | Un | N | 92.2 ± 0.3 |
| B | U | L | Un | G | 99.1 ± 0.4 |
| B | U | N | Un | G | 91.2 ± 0.3 |
| Sp | N ($\delta = 1$) | L | Sp | v | 96.3 ± 0.3 |
| Sp | N ($\delta = 1$) | N | Sp | v | 95.9 ± 0.2 |
| Un | L ($\lambda = 1$) | N | Un | N | 88.5 ± 0.3 |
| Un | N ($\delta = 1$) | N | Un | N | 89.2 ± 0.2 |

| Generative process | | | Model | | |
|---|---|---|---|---|---|
| S | $p(\mathbf{z}_x)$ | $p(\mathbf{z}_x|\mathbf{z}_t)$ | S | $q(\mathbf{z}_x|\mathbf{z}_t)$ | MCC |
| B | U | L | B | L | 99.1 ± 0.1 |
| B | U | G | B | G | 97.2 ± 0.3 |
| B | U | N | B | N | 98.6 ± 0.2 |
| B | U | L | B | N | 99.1 ± 0.1 |
| B | U | G | B | L | 98.4 ± 0.1 |
| B | U | L | Un | L | 95.6 ± 0.2 |
| B | U | G | Un | G | 96.4 ± 0.2 |

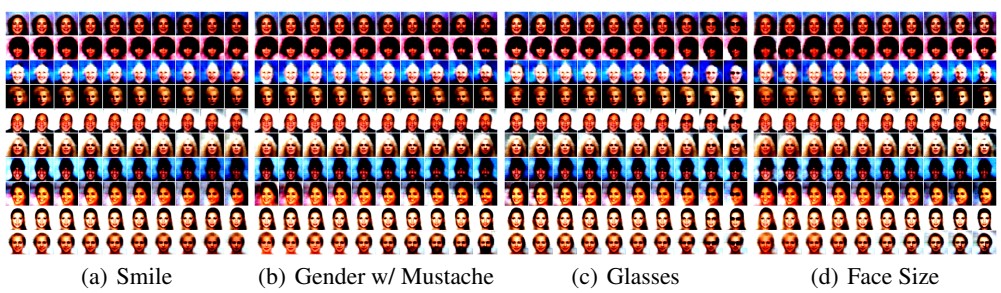

(a) Smile      (b) Gender w/ Mustache      (c) Glasses      (d) Face Size

Figure 3: Disentangled Representations learned by combining pre-trained CLIP and FastICA. The results are aligned with our disentanglement findings.

*experiments, this work validates identifiability through empirical analysis on real datasets and pre-trained CLIP model. This underscores the practicality of our theoretical contributions.*

**Disentangled representations for CelebA data** According to Section 5, we first extract representations from the pre-trained CLIP model and then apply FastICA to these representations to achieve final representations for CelebA data (Liu et al., 2015). We expect these final representations to exhibit clear signs of disentanglement. To validate this, we proceed to train a decoder that reconstructs observational data using these extracted representations. Figure 3 illustrates the effectiveness of our method through latent space traversals. Specifically, it visualizes changes in reconstructions as we traverse one dimension of the latent space at a time, showcasing 4 out of 16 attributes uncovered by our approach. Additional results are available in Appendix H. This achievement not only validates our identifiability results, but also offers a new research line, i.e., learning disentangled representations by CLIP, or exploring how this disentanglement potential relate to the manipulation of pre-trained vision models, such as diffusion models.

**Few-shot learning and domain generalization** The goal of disentangled representations is to learn representations that transfer easily and robustly to downstream tasks, making them well-suited for few-shot learning and resilient to distribution shifts (Fumero et al., 2023). We thus focus on few-shot learning and domain generalization tasks to further evaluate our disentanglement findings. We extract representations from a limited set of labeled samples using a pre-trained CLIP model, combined with FastICA to align with the hypersphere, and with PCA followed by FastICA to align with convex body. These representations are then used to train a linear classifier. We evaluate the methods on ImageNet (Deng et al., 2009) for few-shot learning and test robustness on ImageNet-V2

Table 2: Quantitative results for 2-shot learning and domain generalization by different methods. ①: Linear Probe, ②: Linear Probe with FastICA, and ③: Linear Probe with PCA and FastICA.

| | | SOURCE | TARGET (IMAGENET-) | | | | |
|---|---|---|---|---|---|---|---|
| ENCODERS | METHODS | IMAGENET | V2 | SKETCH | R | A | AVG. |
| RN50 | ① | 31.95 | 26.48 | 8.41 | 20.74 | 7.44 | 15.77 |
| | ② | 34.06 | 28.74 | 8.37 | 21.72 | 10.15 | 17.25 |
| | ③ | 34.12 | 28.68 | 11.55 | 25.57 | 10.15 | 18.99 |
| RN101 | ① | 37.64 | 31.45 | 13.71 | 31.09 | 11.85 | 20.03 |
| | ② | 39.58 | 33.15 | 13.49 | 30.29 | 14.77 | 22.93 |
| | ③ | 39.86 | 33.58 | 17.93 | 35.48 | 14.20 | 25.29 |
| VIT32 | ① | 38.23 | 32.00 | 16.17 | 33.67 | 12.88 | 23.68 |
| | ② | 40.21 | 33.97 | 16.54 | 34.79 | 15.72 | 25.26 |
| | ③ | 39.34 | 33.44 | 19.02 | 36.98 | 14.69 | 26.03 |
| VIT16 | ① | 44.97 | 38.11 | 22.06 | 43.86 | 25.99 | 32.51 |
| | ② | 45.52 | 39.38 | 22.55 | 45.33 | 30.47 | 34.43 |
| | ③ | 46.57 | 40.66 | 26.67 | 49.69 | 31.48 | 37.13 |

(Recht et al., 2019), ImageNet-Sketch (Wang et al., 2019), ImageNet-R (Hendrycks et al., 2021a), and ImageNet-A (Hendrycks et al., 2021b). Table 2 presents the performance metrics of the proposed methods in few-shot learning (the 'SOURCE' column) and domain generalization (the 'TARGET' columns). Analyzing the data in the 'SOURCE' column reveals that the proposed methods outperform the baseline approach, which trains a linear classifier using representations directly obtained from pre-trained CLIP (i.e., the Linear Probe). This superior performance validates our disentanglement findings. The 'TARGET' column further reinforces the benefits of disentanglement. Refer to Appendix I for more results.

**Leveraging the Disentanglement Potential of CLIP-like Models for Few-Shot Learning** In the final experiments, we demonstrate how the disentanglement potential pushes the boundaries of leveraging pre-trained models, e.g., CLIP. Recent progress shows that pre-trained CLIP's adaptability can be significantly improved with just a few labeled training samples. The key to leveraging pre-trained CLIP for few-shot learning is effectively utilizing its extracted representations from limited labeled data, as Tip-Adapter and Tip-Adapter-F methods proposed in the work (Zhang et al., 2022a). As we claim, leveraging disentangled potential of pre-trained CLIP can enhance performance on tasks that rely on disentangled representations, including few-shot learning. Therefore, rather than using CLIP's raw representations, we apply FastICA to extract disentangled representations for few-shot tasks. This can be implemented in a plug-and-play way. As shown in Figure 4, incor-

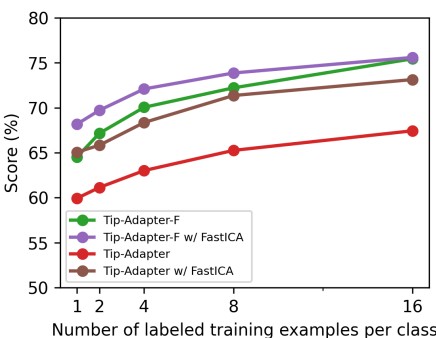

Figure 4: Comparison of accuracy (%) achieved by different few-shot CLIP adaptation methods across 11 datasets.

porating FastICA in the methods in (Zhang et al., 2022a), termed Tip-Adapter with FastICA and Tip-Adapter-F with FastICA, results in better performance, across 11 datasets. See Appendix J for more details.

## 7 CONCLUSION AND DISCUSSIONS

In this work, we propose a novel latent partial causal model for multimodal data that moves beyond the traditional DAG structure, using latent coupled variables connected by undirected edges to capture transferable knowledge across modalities. We establish a theoretical link between this generative

model and MMCL, showing that the representations learned by MMCL correspond to latent variables in the generative model, with linear and permutation transformations in hypersphere and convex body spaces, respectively. Our results provide the first theoretical guarantees for the disentanglement capabilities of MMCL, with applications in tasks like few-shot learning and domain generalization. Unlike prior simulation-based studies, our work demonstrates the real-world utility of MMCL and offers insights into leveraging pre-trained models like CLIP for disentangled representations. Our model challenges conventional DAG assumptions and provides a flexible, practical framework that enhances the effectiveness of MMCL.

One of the main limitations of this work lies in the parametric assumptions, e.g., Eqs. (4) and (7), which may not strictly hold in real-world applications. However, simulation experiments in settings where some of these assumptions are violated indicate that the theoretical results still largely hold (e.g., Table 1). Furthermore, empirical evaluations, including learning disentangled representations on face images (e.g., Figure 3), few-shot learning and domain generalization on ImageNet-type datasets (e.g., Table 2), and few-shot learning across 11 cross-domain datasets (e.g., Figure 4), demonstrate the practical advantages of our theoretical findings, providing additional support for their relevance.

## 8 ACKNOWLEDGMENT

This project was partially funded by the Responsible AI Research Centre (Yuhang Liu, Zhen Zhang, Erdun Gao, Anton van den Hengel, and Javen Qinfeng Shi). Dong Gong was partially supported by the Australian Government through the Australian Research Council Discovery Early Career Researcher Award (DECRA)(DE230101591). Mingming Gong was partially supported by the Australian Government through the Australian Research Council Discovery Projects (DP240102088). The authors also thank the anonymous reviewers for their constructive feedback.

**Ethics Statement.** This study complies with the ethical standards of ICLR. It relies exclusively on public datasets and does not pose foreseeable negative impacts.

**Reproducibility Statement.** We provide thorough descriptions of the methodology and experiments. The code and instructions will be openly available once the paper is published.

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

# Appendix

## Table of Contents

## A   RELATED WORK

**Multimodal contrastive representation learning**   Multi-modal contrastive representation learning, driven by underlying transferable knowledge across modalities, aims to coalesce inputs from these diverse sources into a cohesive representation space. This is typically achieved using a symmetric version of the standard contrastive loss (Oord et al., 2018; Gutmann & Hyvärinen, 2010), a method designed to align accurate pairings while distinguishing incorrect ones (He & Peng, 2017; Radford et al., 2021). Although this approach has proven successful in a range of downstream tasks (Radford et al., 2021; Zhou et al., 2022a;b; Lüddecke & Ecker, 2022; Ban & Dong, 2022), there remains a gap in our comprehensive theoretical and empirical understanding of the representations it learns. Recently, there has been a growing interest in exploring multi-modal contrastive learning from various perspectives. For instance, the study by Liang et al. (2022) provides insights into the modality gap inherent in multi-modal contrastive learning. Similarly, the research presented by Nakada et al. (2023) establishes a link between general multimodal contrastive loss and SVD analysis. Additionally, Huang et al. (2021) posits that learning with multiple modalities can lead to a reduced population risk compared to using a subset of these modalities. Diverging from these approaches, our work delves into multi-modal contrastive representation learning by examining its connection with generative models.

Past research has sought to comprehend the representations derived from standard single-modality contrastive learning, examining them through the lens of alignment and uniformity (Wang & Isola, 2020), showing guarantees on the performance of the learned representations on the average classification task (Saunshi et al., 2019), or in terms of the identifiability of latent variables (Zimmermann et al., 2021; Von Kügelgen et al., 2021). Building on these foundations, our work takes a foreword step. We demonstrate that multi-modal contrastive learning can identify latent coupled variables, extending the insights from previous studies into the realm of multi-modality. Refer to Section G for more details.

Very recently, several studies have emerged, focusing on multi-modal settings (Daunhawer et al., 2023; Yao et al., 2023). A clear distinction is that: the proposed model captures transferable knowledge across modalities by an undirected edge between latent coupled variables, while previous works often achieve it by introducing shared variables (Daunhawer et al., 2023; Yao et al., 2023). Notably, our modeling approach is more general, as it can be reduced to the shared variables used in previous works (Daunhawer et al., 2023; Yao et al., 2023) by enforcing an identical mapping on the undirected edge between latent coupled variables. Some of these works have only achieved partial identifiability of coupled variables (Daunhawer et al., 2023; Yao et al., 2023), specifically identifying latent content variables but not latent style variables. In contrast, our work achieves comprehensive identifiability results for all latent coupled variables, offering a deeper level of understanding. Our research also diverges from the approach taken in Gresele et al. (2020) in two key ways: Firstly, we model ransferable knowledge across modalities using conditional distributions, whereas the latter utilizes identical variables for this purpose. Secondly, while Gresele et al. (2020) relies on the premise that the mapping from the latent space to observations must be constrained by component-wise corrupters to ensure identifiability, our findings do not necessitate such constraints. Refer to Section F for details.

**Nonlinear ICA and Causal Representation Learning**   Nonlinear Independent Component Analysis (ICA) aims to unravel latent independent variables from observational data that has been subject to a nonlinear mixture of these latent factors. However, as pointed out in the seminal work by Hyvärinen & Pajunen (1999), solving this problem is generally infeasible without specific underlying assumptions. A prominent direction in contemporary research leverages the concept of distributional changes in latent variables, which leads to the creation of multi-domain observational data. This approach has been extensively explored and developed in a series of studies (Hyvarinen & Morioka, 2016; 2017; Hyvarinen et al., 2019; Khemakhem et al., 2020), each contributing to a deeper understanding and more refined methodologies in the field of Nonlinear ICA. We build upon this body of research by incorporating co-occurrence patterns observed across multiple modalities. It is important to note the distinct difference between multi-domain and multi-modal approaches. The former typically implies a consistent mapping from the latent space to the observational space across all domains, whereas the latter accommodates different mappings for each modality. Additionally, while multi-domain approaches generally assume a totally shared latent variables across all domains, multi-modal methods allow for the existence of modality-specific latent variables. This work is also

distinct from prior studies on causal representation learning (Von Kügelgen et al., 2021; Liu et al., 2022; Buchholz et al., 2023; Liu et al., 2024b; 2025; 2024a; 2026; Brehmer et al., 2022; Ahuja et al., 2023; Seigal et al., 2022; Varici et al., 2023). While these works aim to address more general causal settings, achieving identifiability in such regimes typically requires substantially stronger assumptions, which can limit their applicability in real-world scenarios. In contrast, our approach is empirically grounded: extensive experiments with pre-trained CLIP models across a diverse set of real-world tasks demonstrate practical benefits.

## B  THE PROOF OF THEOREM 3.1

**Theorem 3.1.** *(The asymptotics of $\mathcal{L}$) For fixed $\tau > 0$, as the sample size $N \to \infty$, the (normalized) multimodal contrastiveloss converges to*

$$\lim_{N \to \infty} \mathcal{L} - 2\log N = 2 \underset{(\mathbf{x},\mathbf{t}) \sim p(\mathbf{x},\mathbf{t})}{\mathbb{E}} \left[ d\big(\mathbf{f}_x(\mathbf{x}), \mathbf{f}_t(\mathbf{t})\big)/\tau \right] + \underset{\mathbf{x} \sim p(\mathbf{x})}{\mathbb{E}} \left[ \log \underset{\mathbf{t} \sim p(\mathbf{t})}{\mathbb{E}} \left[ e^{-d\big(\mathbf{f}_x(\mathbf{x}), \mathbf{f}_t(\mathbf{t})\big)/\tau} \right] \right] \quad (10)$$

$$+ \underset{\mathbf{t} \sim p(\mathbf{t})}{\mathbb{E}} \left[ \log \underset{\mathbf{x} \sim p(\mathbf{x})}{\mathbb{E}} \left[ e^{-d\big(\mathbf{f}_x(\mathbf{x}), \mathbf{f}_t(\mathbf{t})\big)/\tau} \right] \right].$$

*Proof.* This proof is done by mainly depending on the Continuous Mapping Theorem and the law of large numbers.

$$\lim_{N \to \infty} \mathcal{L} - 2\log N = \lim_{N \to \infty} \left( -\frac{1}{N} \sum_{i=1}^{N} \log \frac{e^{-d\big(\mathbf{f}_x(\mathbf{x}_i), \mathbf{f}_t(\mathbf{t}_i)\big)/\tau}}{\sum_{j=1}^{N} e^{-d\big(\mathbf{f}_x(\mathbf{x}_i), \mathbf{f}_t(\mathbf{t}_j)\big)/\tau}} \right.$$

$$\left. -\frac{1}{N} \sum_{i=1}^{N} \log \frac{e^{-d\big(\mathbf{f}_x(\mathbf{x}_i), \mathbf{f}_t(\mathbf{t}_i)\big)/\tau}}{\sum_{j=1}^{N} e^{-d\big(\mathbf{f}_x(\mathbf{x}_j), \mathbf{f}_t(\mathbf{t}_i)\big)/\tau}} \right) - 2\log N,$$

$$= \lim_{N \to \infty} \left( \frac{2}{N} \sum_{i=1}^{N} d\big(\mathbf{f}_x(\mathbf{x}_i), \mathbf{f}_t(\mathbf{t}_i)\big)/\tau + \frac{1}{N} \sum_{i=1}^{N} \log \sum_{j=1}^{N} e^{-d\big(\mathbf{f}_x(\mathbf{x}_i), \mathbf{f}_t(\mathbf{t}_j)\big)/\tau} \right.$$

$$\left. + \frac{1}{N} \sum_{i=1}^{N} \log \sum_{j=1}^{N} e^{-d\big(\mathbf{f}_x(\mathbf{x}_j), \mathbf{f}_t(\mathbf{t}_i)\big)/\tau} \right) - 2\log N$$

$$= \lim_{N \to \infty} \left( \frac{2}{N} \sum_{i=1}^{N} d\big(\mathbf{f}_x(\mathbf{x}_i), \mathbf{f}_t(\mathbf{t}_i)\big)/\tau + \frac{1}{N} \sum_{i=1}^{N} \log \frac{1}{N} \sum_{j=1}^{N} e^{-d\big(\mathbf{f}_x(\mathbf{x}_i), \mathbf{f}_t(\mathbf{t}_j)\big)/\tau} \right.$$

$$\left. + \frac{1}{N} \sum_{i=1}^{N} \log \frac{1}{N} \sum_{j=1}^{N} e^{-d\big(\mathbf{f}_x(\mathbf{x}_j), \mathbf{f}_t(\mathbf{t}_i)\big)/\tau} + \frac{2}{N} \sum_{i=1}^{N} \log N \right) - 2\log N$$

$$= 2 \underset{(\mathbf{x},\mathbf{t}) \sim p(\mathbf{x},\mathbf{t})}{\mathbb{E}} \left[ d\big(\mathbf{f}_x(\mathbf{x}), \mathbf{f}_t(\mathbf{t})\big)/\tau \right] + \underset{\mathbf{x} \sim p(\mathbf{x})}{\mathbb{E}} \left[ \log \underset{\mathbf{t} \sim p(\mathbf{t})}{\mathbb{E}} \left[ e^{-d\big(\mathbf{f}_x(\mathbf{x}), \mathbf{f}_t(\mathbf{t})\big)/\tau} \right] \right]$$

$$+ \underset{\mathbf{t} \sim p(\mathbf{t})}{\mathbb{E}} \left[ \log \underset{\mathbf{x} \sim p(\mathbf{x})}{\mathbb{E}} \left[ e^{-d\big(\mathbf{f}_x(\mathbf{x}), \mathbf{f}_t(\mathbf{t})\big)/\tau} \right] \right].$$

$\square$

## C  RELATION WITH RECOVERING ALL INFORMATION

In this section, we show

$$\underset{\mathbf{x} \sim p(\mathbf{x})}{\mathbb{E}} \left[ \log \underset{\mathbf{t} \sim p(\mathbf{t})}{\mathbb{E}} \left[ e^{-d\big(\mathbf{f}_x(\mathbf{x}), \mathbf{f}_t(\mathbf{t})\big)/\tau} \right] \right] + \underset{\mathbf{t} \sim p(\mathbf{t})}{\mathbb{E}} \left[ \log \underset{\mathbf{x} \sim p(\mathbf{x})}{\mathbb{E}} \left[ e^{-d\big(\mathbf{f}_x(\mathbf{x}), \mathbf{f}_t(\mathbf{t})\big)/\tau} \right] \right]$$

$$\approx - H\big(p(\mathbf{f}_x(\mathbf{x})), p(\mathbf{f}_t(\mathbf{t}))\big) - H\big(p(\mathbf{f}_t(\mathbf{t})), p(\mathbf{f}_x(\mathbf{x}))\big).$$

Considering the symmetry evident in both the left and right sides of the equation, let us focus our attention on the initial term on the left and its corresponding counterpart on the right.

$$\mathbb{E}_{\mathbf{x} \sim p(\mathbf{x})} \left[ \log \mathbb{E}_{\mathbf{t} \sim p(\mathbf{t})} \left[ e^{-d\left(\mathbf{f}_x(\mathbf{x}), \mathbf{f}_t(\mathbf{t})\right)/\tau} \right] \right]$$

$$= \lim_{N \to \infty} \frac{1}{N} \sum_{i=1}^{N} \log \frac{1}{N} \sum_{j=1}^{N} e^{-d\left(\mathbf{f}_x(\mathbf{x}_i), \mathbf{f}_t(\mathbf{t}_j)\right)/\tau} \tag{11}$$

$$\approx \lim_{N \to \infty} \frac{1}{N} \sum_{i=1}^{N} \log p_{\text{KDE}}(\mathbf{f}_x(\mathbf{x}_i)) + \log Z_{\text{KDE}} \tag{12}$$

$$= - H(p(\mathbf{f}_x(\mathbf{x}), p(\mathbf{f}_t(\mathbf{t}))) + \log Z_{\text{KDE}}, \tag{13}$$

Transitioning from Eq. (11) to Eq. (12), we employ kernel density estimation, wherein the choice of kernel is influenced by the distance metric used. For instance, on a hypersphere, a von Mises-Fisher kernel is suitable, whereas on convex bodies, a Laplace kernel aligns well with the L1 norm. In this context, $\log Z_{\text{KDE}}$ represents the normalization constant associated with the kernel. The inherent symmetry in this setup allows us to logically deduce the equation. Note that since here the bandwidth $\tau$ can be optimized in MMCL, if true distribution is the same as the chosen kernel, Eq. (12) is equal to Eq. (11), *i.e.*, $\approx$ in Eq. (12) can be =. Under certain conditions the kernel density estimation will converge to the real distribution, in that case $\approx$ in Eq. (12) can also be = (Silverman, 2018). Specifically, kernel density estimation converges to the true density $p(\mathbf{f}_x(\mathbf{x}))$ under the following assumptions:

- The kernel $K_\tau(\mathbf{z}, \mathbf{z}') = e^{-d(\mathbf{z},\mathbf{z}')/\tau}$ is a smooth, symmetric density function (e.g., Gaussian-like).
- The number of samples $N \to \infty$, and the bandwidth $\tau \to 0$, such that $N\tau^d \to \infty$.
- The true density $p(\mathbf{f}_t(\mathbf{t}))$ is smooth (e.g., twice differentiable with bounded second derivatives).

These conditions ensure that $p_{\text{KDE}}(\mathbf{f}_x(\mathbf{x}_i))$ converges to the true density.

## D  THE PROOF OF IDENTIFIABILITY ON HYPERSPHERE

### D.1  THE PROOF OF THEOREM 4.1

**Theorem 4.1.** *($\mathcal{L}$ converges to the symmetric cross-entropy) Under the assumptions defined in Eq. (4) for the proposed latent partial causal model, the necessary condition $\mathbf{f}_x \circ \mathbf{g}_x = \mathbf{f}_t \circ \mathbf{g}_t$, denoted as $\mathbf{h}$, for the optimal normalized multimodal contrastive loss given by Eq. (2) leads to the following reduction of the loss itself:*

$$\lim_{N \to \infty} \mathcal{L} - 2 \log N = \mathbb{E}_{\mathbf{z}_x \sim p(\mathbf{z}_x)} \left[ H(p(\mathbf{z}_t|\mathbf{z}_x), q_{\mathbf{h}}(\mathbf{z}_t|\mathbf{z}_x)) \right] + \mathbb{E}_{\mathbf{z}_t \sim p(\mathbf{z}_t)} \left[ H(p(\mathbf{z}_x|\mathbf{z}_t), q_{\mathbf{h}}(\mathbf{z}_x|\mathbf{z}_t)) \right] \tag{14}$$

*where $H$ is the cross entropy, the conditional distributions $q_{\mathbf{h}}(\mathbf{z}_t|\mathbf{z}_x)$ and $q(\mathbf{z}_x|\mathbf{z}_t)$ are parameterized by the following:*

$$q_{\mathbf{h}}(\mathbf{z}_x|\mathbf{z}_t) = C_q(\mathbf{z}_t)^{-1} e^{(\mathbf{h}(\mathbf{z}_x)^T \mathbf{h}(\mathbf{z}_t)/\tau)}, \tag{15}$$

$$q_{\mathbf{h}}(\mathbf{z}_t|\mathbf{z}_x) = C_q(\mathbf{z}_x)^{-1} e^{(\mathbf{h}(\mathbf{z}_t)^T \mathbf{h}(\mathbf{z}_x)/\tau)}, \tag{16}$$

*with*

$$C_q(\mathbf{z}_t) = \int e^{(\mathbf{h}(\mathbf{z}_x)^T \mathbf{h}(\mathbf{z}_t)/\tau)} \mathrm{d}\mathbf{z}_x,$$

$$C_q(\mathbf{z}_x) = \int e^{(\mathbf{h}(\mathbf{z}_x)^T \mathbf{h}(\mathbf{z}_t)/\tau)} \mathrm{d}\mathbf{z}_t.$$

To proof Theorem 4.1, we first introduce the following Lemma.

**Lemma 1.** *Consider the unit hypersphere space, given uniform prior $p(\mathbf{z}_x)$, $p(\mathbf{z}_x) = |\mathcal{Z}|^{-1}$ where $\mathcal{Z} \subseteq \mathbb{R}^M$ denotes the space of $\mathbf{z}_x$, and conditional distribution $p(\mathbf{z}_t|\mathbf{z}_x) = C_p(k) \exp(k\mathbf{z}_x^T \mathbf{z}_t)$, $p(\mathbf{z}_t)$ follows a uniform distribution.*

*Proof.* By Bayesian theorem, $p(\mathbf{z}_t) = \int p(\mathbf{z}_t|\mathbf{z}_x)p(\mathbf{z}_x)\mathrm{d}\mathbf{z}_x = |\mathcal{Z}|^{-1}\int p(\mathbf{z}_t|\mathbf{z}_x)\mathrm{d}\mathbf{z}_x = |\mathcal{Z}|^{-1}C_p(k)\int \exp\left(k\mathbf{z}_x^T\mathbf{z}_t\right)\mathrm{d}\mathbf{z}_x$, then due to the unit hypersphere space, we have $\int \exp\left(k\mathbf{z}_x^T\mathbf{z}_t\right)\mathrm{d}\mathbf{z}_x = C_p(k)^{-1}$. As a result, we obtain $p(\mathbf{z}_t) = |\mathcal{Z}|^{-1}$. $\square$

**Lemma 2.** *The normalized multimodal contrastive loss in Eq.* (2) *has an optimal global solution of 0, which can be attained under the following conditions:*

- $\mathbf{h}_x(\mathbf{m}_x, \mathbf{z}_x) = \mathbf{h}_t(\mathbf{m}_t, \mathbf{z}_t)$ *almost surely, for pair* $\left((\mathbf{m}_x, \mathbf{z}_x), (\mathbf{m}_t, \mathbf{z}_t)\right)$, *(C1),*

- $\mathbf{h}_x$ *and* $\mathbf{h}_t$ *map* $(\mathbf{m}_x, \mathbf{z}_x)$ *and* $(\mathbf{m}_t, \mathbf{z}_t)$*, respectively, to uniform variables on hypersphere, (C2),*

*Proof.* First, it is well known that traditional contrastive loss in single modality has an optimal global solution of $\log N$ (Oord et al., 2018; Tian et al., 2020), as a result, the multimodal contrastive loss Eq. 1 has an optimal global solution of $2\log N$. For completeness, let us focus on the first term in Eq. 1:

$$-\frac{1}{N}\sum_{i=1}^{N}\log\frac{e^{-d\left(\mathbf{f}_x(\mathbf{x}_i),\mathbf{f}_t(\mathbf{t}_i)\right)/\tau}}{\sum_{j=1}^{N}e^{-d\left(\mathbf{f}_x(\mathbf{x}_i),\mathbf{f}_t(\mathbf{t}_j)\right)/\tau}}, \tag{17}$$

Under optimal contrastive learning conditions, the distance for positive pairs satisfies: $e^{-d\left(\mathbf{f}_x(\mathbf{x}_i),\mathbf{f}_t(\mathbf{t}_i)\right)/\tau} = 1$, for negative pairs $(\mathbf{x}_i, \mathbf{x}_j)$ where $i \neq j$: $e^{-d\left(\mathbf{f}_x(\mathbf{x}_i),\mathbf{f}_t(\mathbf{t}_j)\right)/\tau} = \epsilon$, where $\epsilon$ is a small value. As a result, for each $i$, the denominator can be expressed as: $1 + (N-1)\epsilon$. Therefore, the first term in Eq. 1 reduces to : $-\frac{1}{N}\sum_{i=1}^{N}\log\frac{1}{1+(N-1)\epsilon}$. Clearly, when $N$ is large, the first term in Eq. 1 equals to $\log N$. Given that the second term is symmetric, we conclude that Eq. 1 has an optimal global solution of $2\log N$. Therefore, Eq. 10 achieves a global optimal solution of 0. Moreover, this optimum is unique up to isometries and permutations. Minimizing the loss requires each positive pair to dominate its softmax denominator, which is only achieved when their embeddings are maximally aligned. Simultaneously, negative pairs must be mapped as far apart as possible under the bounded metric to minimize their influence. This configuration, tight positive alignment and maximal negative separation, is geometrically rigid: any deviation increases the loss. Thus, except for distance-preserving transformations and index permutations, the solution is unique. Achieving the global minimum of Eq. 10 therefore necessitates maximizing the alignment of positive pairs. This occurs if and only if $\mathbf{h}_x(\mathbf{m}_x, \mathbf{z}_x) = \mathbf{h}_t(\mathbf{m}_t, \mathbf{z}_t)$ almost surely, for real pair $\left((\mathbf{m}_x, \mathbf{z}_x), (\mathbf{m}_t, \mathbf{z}_t)\right)$, (marked as (C1)). Thus, we obtain a minimum solution of 0 for the first term. Next, considering the remaining two terms in Eq. 10, as detailed in Appendix C, we see an equivalent expression: $-H(p(\mathbf{f}_x(\mathbf{x}), p(\mathbf{f}_t(\mathbf{t}))) - H(p(\mathbf{f}_x(\mathbf{x}), p(\mathbf{f}_t(\mathbf{t}))) + 2\log Z_{\mathrm{KDE}}$. When both $\mathbf{h}_x$ and $\mathbf{h}_t$ map $(\mathbf{m}_x, \mathbf{z}_x)$ and $(\mathbf{m}_t, \mathbf{z}_t)$, respectively, to uniform variables on hypersphere (marked as (C2)), it reduces to $-2H(p(\mathbf{f}_x(\mathbf{x})) + 2\log Z_{\mathrm{KDE}}$. Note that the entropy of a uniform distribution on the hypersphere $\mathbb{S}^{M-1}$ is $\log(\frac{2\pi^{M/2}}{\Gamma(M/2)})$, where $\Gamma$ is the gamma function. Together with the fact that the normalization constant of uniform distribution on hypersphere is $\log(\frac{2\pi^{M/2}}{\Gamma(M/2)})$ (i.e., $\log Z_{\mathrm{KDE}}$), we arrive at the optimal solution of 0 for the last two terms. $\square$

**Proof sketch** The proof of Theorem 4.1 hinges on demonstrating the equality between the right-hand side of Eq. (14) and Eq. (10). Let us define $\mathbf{h}_x = \mathbf{f}_x \circ \mathbf{g}_x$ and $\mathbf{h}_t = \mathbf{f}_t \circ \mathbf{g}_t$. In Step I, using Lemma 2, we show that (1) $\mathbf{f}_x \circ \mathbf{g}_x = \mathbf{f}_t \circ \mathbf{g}_t$, and (2) they are independent of the modality-specific variables $\mathbf{m}_x$ and $\mathbf{m}_t$. In Step II, by defining $\mathbf{h} = \mathbf{f}_x \circ \mathbf{g}_x = \mathbf{f}_t \circ \mathbf{g}_t$ and applying both the generative model from Eq. (4) and the inference model from Eqs. (15)-(16), we establish the theorem.

**Step I** Consider C1 in Lemma 2, e.g., $\mathbf{h}_x(\mathbf{m}_x, \mathbf{z}_x) = \mathbf{h}_t(\mathbf{m}_t, \mathbf{z}_t)$ almost surely, for pair $\left((\mathbf{m}_x, \mathbf{z}_x), (\mathbf{m}_t, \mathbf{z}_t)\right)$, by differentiating it with respect to $\mathbf{m}_x$, we have:

$$\frac{\partial \mathbf{h}_x(\mathbf{m}_x, \mathbf{z}_x)}{\partial \mathbf{m}_x} = \frac{\partial \mathbf{h}_t(\mathbf{m}_t, \mathbf{z}_t)}{\partial \mathbf{m}_x} = 0, \tag{18}$$

, due to the independence between $\mathbf{m}_x$ and $(\mathbf{m}_t, \mathbf{z}_t)$. Similarly, by differentiating it with respect to $\mathbf{m}_t$, we have:

$$\frac{\partial \mathbf{h}_t(\mathbf{m}_t, \mathbf{z}_t)}{\partial \mathbf{m}_t} = \frac{\partial \mathbf{h}_x(\mathbf{m}_x, \mathbf{z}_x)}{\partial \mathbf{m}_t} = 0. \tag{19}$$

Based on Eqs. (18) and (19), we conclude that both $\mathbf{h}_x$ and $\mathbf{h}_t$ are independent of the modality-specific variables $\mathbf{m}_x$ and $\mathbf{m}_t$, respectively, *i.e.*, $\mathbf{h}_x(\mathbf{m}_x, \mathbf{z}_x) = \mathbf{h}_x(\mathbf{z}_x)$ and $\mathbf{h}_t(\mathbf{m}_t, \mathbf{z}_t) = \mathbf{h}_t(\mathbf{z}_t)$. As a result, we have $\mathbf{h}_x(\mathbf{z}_x) = \mathbf{h}_t(\mathbf{z}_t)$, for all real pairs $(\mathbf{z}_x, \mathbf{z}_t)$ sampled from the conditional distribution $p(\mathbf{z}_t|\mathbf{z}_x)$ defined in Eq. (4). Note that this expression also holds true for $\mathbf{z}_t = \mathbf{z}_x$ (e.g., when $\mathbf{z}_t$ is sampled with the same value as $\mathbf{z}_x$), which implies $\mathbf{h}_x(\mathbf{z}_x) = \mathbf{h}_t(\mathbf{z}_x)$. As a result, we can obtain: $\mathbf{h}_x = \mathbf{h}_t$.

**Step II**  According to the results above: $\mathbf{h}_x(\mathbf{m}_x, \mathbf{z}_x) = \mathbf{h}_x(\mathbf{z}_x)$, $\mathbf{h}_t(\mathbf{m}_t, \mathbf{z}_t) = \mathbf{h}_t(\mathbf{z}_x)$, and $\mathbf{h}_x = \mathbf{h}_t$ from Step I, by defining $\mathbf{h} \overset{\text{def}}{=} \mathbf{h}_x = \mathbf{h}_t$ , we can rewrite Eq. (10) as:

$$2 \mathop{\mathbb{E}}_{(\mathbf{z}_x, \mathbf{z}_t) \sim p(\mathbf{z}_x, \mathbf{z}_t)} \left[ d\big(\mathbf{h}(\mathbf{z}_x), \mathbf{h}(\mathbf{z}_t)\big)/\tau \right] + \mathop{\mathbb{E}}_{\mathbf{z}_x \sim p(\mathbf{z}_x)} \left[ \log \mathop{\mathbb{E}}_{\mathbf{z}_t \sim p(\mathbf{z}_t)} \left[ e^{-d\big(\mathbf{h}(\mathbf{z}_x), \mathbf{h}(\mathbf{z}_t)\big)/\tau} \right] \right]$$

$$+ \mathop{\mathbb{E}}_{\mathbf{z}_t \sim p(\mathbf{z}_t)} \left[ \log \mathop{\mathbb{E}}_{\mathbf{z}_x \sim p(\mathbf{z}_x)} \left[ e^{-d\big(\mathbf{h}(\mathbf{z}_x), \mathbf{h}(\mathbf{z}_t)\big)/\tau} \right] \right]. \tag{20}$$

We then connect the right-hand side of Eq. (14) with Eq. (20). To this end, since the two terms in the right-hand side of Eq. (14) are symmetrical, we focus on one of the two terms for convenience, *e.g.*, $\mathop{\mathbb{E}}_{\mathbf{z}_x \sim p(\mathbf{z}_x)} \left[ H(p(\mathbf{z}_t|\mathbf{z}_x)), q_{\mathbf{h}}(\mathbf{z}_t|\mathbf{z}_x)) \right]$. Based on Lemma 1, it can be shown that:

$$\mathop{\mathbb{E}}_{\mathbf{z}_x \sim p(\mathbf{z}_x)} \left[ H(p(\mathbf{z}_t|\mathbf{z}_x)), q_{\mathbf{h}}(\mathbf{z}_t|\mathbf{z}_x)) \right] \tag{21}$$

$$= \mathop{\mathbb{E}}_{\mathbf{z}_x \sim p(\mathbf{z}_x)} \left[ \mathop{\mathbb{E}}_{\mathbf{z}_t \sim p(\mathbf{z}_t|\mathbf{z}_x)} \left[ -\log q_{\mathbf{h}}(\mathbf{z}_t|\mathbf{z}_x) \right] \right] \tag{22}$$

$$= \mathop{\mathbb{E}}_{(\mathbf{z}_x, \mathbf{z}_t) \sim p(\mathbf{z}_x, \mathbf{z}_t)} \left[ -\mathbf{h}(\mathbf{z}_x)^T \mathbf{h}(\mathbf{z}_t)/\tau + \log C_q(\mathbf{z}_x) \right] \tag{23}$$

$$= \mathop{\mathbb{E}}_{(\mathbf{z}_x, \mathbf{z}_t) \sim p(\mathbf{z}_x, \mathbf{z}_t)} \left[ -\mathbf{h}(\mathbf{z}_x)^T \mathbf{h}(\mathbf{z}_t)/\tau \right] + \mathop{\mathbb{E}}_{(\mathbf{z}_x) \sim p(\mathbf{z}_x)} \left[ \log C_q(\mathbf{z}_x) \right] \tag{24}$$

$$= \mathop{\mathbb{E}}_{(\mathbf{z}_x, \mathbf{z}_t) \sim p(\mathbf{z}_x, \mathbf{z}_t)} \left[ -\mathbf{h}(\mathbf{z}_x)^T \mathbf{h}(\mathbf{z}_t)/\tau \right] + \mathop{\mathbb{E}}_{(\mathbf{z}_x) \sim p(\mathbf{z}_x)} \left[ \log \int e^{(\mathbf{h}(\mathbf{z}_x)^T \mathbf{h}(\mathbf{z}_t)/\tau)} \mathrm{d}\mathbf{z}_x \right] \tag{25}$$

Since $p(\mathbf{z}_x) = |\mathcal{Z}|^{-1}$, and $p(\mathbf{z}_t) = |\mathcal{Z}|^{-1}$ by Lemma 1, Eq. (25) simplifies to:

$$= -\mathop{\mathbb{E}}_{(\mathbf{z}_x, \mathbf{z}_t) \sim p(\mathbf{z}_x, \mathbf{z}_t)} \left[ (\mathbf{h}(\mathbf{z}_x)^T \mathbf{h}(\mathbf{z}_t))/\tau \right] + \mathop{\mathbb{E}}_{\mathbf{z}_x \sim p(\mathbf{z}_x)} \left[ \log \mathop{\mathbb{E}}_{\mathbf{z}_t \sim p(\mathbf{z}_t)} \left[ e^{(\mathbf{h}(\mathbf{z}_x)^T \mathbf{h}(\mathbf{z}_t))/\tau} \right] \right] + \log |\mathcal{Z}| \tag{26}$$

On hypersphere space with radius $r$, due to $\|\mathbf{h}(\mathbf{z}_x) - \mathbf{h}(\mathbf{z}_t)\| = 2r - 2\mathbf{h}(\mathbf{z}_x)^T \mathbf{h}(\mathbf{z}_t)$, Eq. 26 simplifies to:

$$= \mathop{\mathbb{E}}_{(\mathbf{z}_x, \mathbf{z}_t) \sim p(\mathbf{z}_x, \mathbf{z}_t)} \left[ d\big(\mathbf{h}(\mathbf{z}_x), \mathbf{h}(\mathbf{z}_t)\big)/\tau \right] + \mathop{\mathbb{E}}_{\mathbf{z}_x \sim p(\mathbf{z}_x)} \left[ \log \mathop{\mathbb{E}}_{\mathbf{z}_t \sim p(\mathbf{z}_t)} \left[ e^{-d\big(\mathbf{h}(\mathbf{z}_x)\mathbf{h}(\mathbf{z}_t)\big)/\tau} \right] \right] \tag{27}$$

Similarly, for the second term in the right-hand side of Eq. (14), we can proof that:

$$\mathop{\mathbb{E}}_{(\mathbf{z}_t) \sim p(\mathbf{z}_t)} \left[ H(p(\mathbf{z}_x|\mathbf{z}_t)), q_{\mathbf{h}}(\mathbf{z}_x|\mathbf{z}_t)) \right] = \mathop{\mathbb{E}}_{(\mathbf{z}_x, \mathbf{z}_t) \sim p(\mathbf{z}_x, \mathbf{z}_t)} \left[ d\big(\mathbf{h}(\mathbf{z}_x), \mathbf{h}(\mathbf{z}_t)\big)/\tau \right]$$

$$+ \mathop{\mathbb{E}}_{\mathbf{z}_t \sim p(\mathbf{z}_t)} \left[ \log \mathop{\mathbb{E}}_{\mathbf{z}_x \sim p(\mathbf{z}_x)} \left[ e^{-d\big(\mathbf{h}(\mathbf{z}_x), \mathbf{h}(\mathbf{z}_t)\big)/\tau} \right] \right] + \log |\mathcal{Z}|. \tag{28}$$

By combining Eq. (27) and Eq. (28), we can conclude the proof.

### D.2  IDENTIFIABILITY RESULT ON HYPERSPHERE

Theorem 4.1 represents a adaptation of Theorem 1 from (Zimmermann et al., 2021) in the context of multi-modal setting. Specifically, within the confines of a single-modal framework, Theorem 4.1 is consistent with the findings presented in Theorem 1 in (Zimmermann et al., 2021). Consequently, this alignment allows us to employ Propositions 1 and 2 from (Zimmermann et al., 2021) to demonstrate that the global minimization of the objective outlined in Eq. (5), as specified in Theorem 4.1, identifies

the latent variables $\mathbf{z}_x$, as well as $\mathbf{z}_x$, up to linear transformations. For completeness, a brief proof is provided herein, with comprehensive details available in the original work. Clearly, the global minimum of the cross-entropy between two distributions is reached if they match by value and have the same support. Therefore, for the optimal solution of the objective loss Eq. (14) in Theorem 4.1, we have:

$$p(\mathbf{z}_t|\mathbf{z}_x) = q_\mathbf{h}(\mathbf{z}_t|\mathbf{z}_x), \tag{29}$$

This expression also holds true for $\mathbf{z}_t = \mathbf{z}_x$; additionally using that $\mathbf{h}$ maps from a unit hypersphere to one with radius $\sqrt{\tau k}$, we have:

$$C_p^{-1} e^{(k\mathbf{z}_x^T \mathbf{z}_x)} = C_q(\mathbf{z}_x)^{-1} e^{(\mathbf{h}(\mathbf{z}_x)^T \mathbf{h}(\mathbf{z}_x)/\tau)},$$
$$\Leftrightarrow C_p = C_q(\mathbf{z}_x) \tag{30}$$

As the normalization constants are identical we get for all $\mathbf{z}_x, \mathbf{z}_t$,

$$k\mathbf{z}_x^T \mathbf{z}_t = \mathbf{h}(\mathbf{z}_x)^T \mathbf{h}(\mathbf{z}_t)/\tau, \tag{31}$$

here we can see that $\mathbf{h}$ maintains the dot product, which implies that $\mathbf{h}$ must be an orthogonal linear transformation by using Proposition 2 in Zimmermann et al. (2021). As a result, Theorem 4.1 supports the conclusion that the latent variables $\mathbf{z}_x$ (and $\mathbf{z}_t$) can be identified up to an orthogonal linear transformation, *i.e.*, the recovered latent variables $\mathbf{f}_x(\mathbf{x})$ (note that $\mathbf{h}(\mathbf{z}_x) = \mathbf{f_x}(\mathbf{x})$), obtained by the minimization of Eq. (5), is linearly related to the true $\mathbf{z}_x$ as follows: $\mathbf{f}_x(\mathbf{x}) = \mathbf{A}\mathbf{z}_x + \mathbf{c}$, where $\mathbf{A}$ represents an orthogonal matrix, and $\mathbf{c}$ is a constant vector.

## E    THE PROOF OF IDENTIFIABILITY ON CONVEX BODIES

### E.1    THE PROOF OF THEOREM 4.2

**Theorem 4.2.** *($\mathcal{L}$ converges to the symmetric cross-entropy) Under the assumptions defined in Eq. (7) for the proposed latent partial causal model, the necessary condition $\mathbf{f}_x \circ \mathbf{g}_x = \mathbf{f}_t \circ \mathbf{g}_t$, denoted as $\mathbf{h}$, for the optimal normalized multimodal contrastiveloss given by Eq. (2) leads to the following reduction of the loss itself:*

$$\lim_{N \to \infty} \mathcal{L} - 2\log N = \underset{\mathbf{z}_x \sim p(\mathbf{z}_x)}{\mathbb{E}} \left[ H(p(\mathbf{z}_t|\mathbf{z}_x)), q_\mathbf{h}(\mathbf{z}_t|\mathbf{z}_x)) \right] + \underset{(\mathbf{z}_t) \sim p(\mathbf{z}_t)}{\mathbb{E}} \left[ H(p(\mathbf{z}_x|\mathbf{z}_t)), q_\mathbf{h}(\mathbf{z}_x|\mathbf{z}_t)) \right] \tag{32}$$

*where $H$ is the cross entropy, the conditional distributions $q_\mathbf{h}(\mathbf{z}_t|\mathbf{z}_x)$ and $q(\mathbf{z}_x|\mathbf{z}_t)$ are parameterized by the following:*

$$q_\mathbf{h}(\mathbf{z}_x|\mathbf{z}_t) = C_q(\mathbf{z}_t)^{-1} e^{-\delta(\mathbf{h}(\mathbf{z}_x), \mathbf{h}(\mathbf{z}_t))/\tau}, \tag{33}$$

$$q_\mathbf{h}(\mathbf{z}_t|\mathbf{z}_x) = C_q(\mathbf{z}_x)^{-1} e^{-\delta(\mathbf{h}(\mathbf{z}_x), \mathbf{h}(\mathbf{z}_t))/\tau}, \tag{34}$$

*with*

$$C_q(\mathbf{z}_t) = \int e^{-\delta(\mathbf{h}(\mathbf{z}_x), \mathbf{h}(\mathbf{z}_t))/\tau} d\mathbf{z}_x,$$

$$C_q(\mathbf{z}_x) = \int e^{-\delta(\mathbf{h}(\mathbf{z}_x), \mathbf{h}(\mathbf{z}_t))/\tau} d\mathbf{z}_t.$$

Similar to the proof D.1, we first introduce the following Lemma.

**Lemma 3.** *For random variables $\mathbf{z}_x \in \mathcal{Z}_c$ and $\mathbf{z}_t = \mathcal{Z}_c$, assume that $p(\mathbf{z}_x) = 1/|\mathcal{Z}_c|$ if $\mathbf{z}_x \in \mathcal{Z}_c$ and $0$ otherwise, and assume that conditional distribution $p(\mathbf{z}_t|\mathbf{z}_x) = C(\mathbf{z}_x) \exp\left(-\delta(\mathbf{z}_x, \mathbf{z}_t)/\lambda\right)$, where $\delta$ is a symmetric metric induced by a norm, then $p(\mathbf{z}_t)$ converges to uniform distribution on $\mathcal{Z}_c$ as $\lambda \to 0_+$.*

*Proof.* The proof can be done by the fact that as $\lambda \to 0$, the condition distribution $p(\mathbf{z}_t|\mathbf{z}_x)$ converges to a delta distribution, resulting that $p(\mathbf{z}_t) = p(\mathbf{z}_x)$. More specifically, as we will let $\lambda \to 0$ in the procedure, it is notable that the normalize $C(\mathbf{z}_x)$ actually depend on $\lambda$ and should be write as

$C(\mathbf{z}_x, \lambda)$ in a more formal way. With simple integration trick, it would be straightforward to show that $C(\mathbf{z}_x, \lambda)$ can be decomposed as $C(\mathbf{z}_x, \lambda) = \frac{1}{\lambda}C'(\mathbf{z}_x)$.

By definition we have

$$
\begin{aligned}
p(\mathbf{z}_t) &= \int_{\mathbf{z}_x \in \mathcal{Z}_c} p(\mathbf{z}_x) p(\mathbf{z}_t | \mathbf{z}_x) \mathrm{d}\mathbf{z}_x \\
&= \int_{\mathbf{z}_x \in \mathcal{Z}_c} p(\mathbf{z}_x) \frac{1}{\lambda} C'(\mathbf{z}_x) \exp\left(-\delta(\mathbf{z}_x, \mathbf{z}_t)/\lambda\right) \mathrm{d}\mathbf{z}_x \\
&= \lim_{N \to +\infty} \sum_{i=1}^{N} \frac{1}{\lambda} C'(\mathbf{z}_{x_i}) \exp\left(-\delta(\mathbf{z}_{x_i}, \mathbf{z}_t)/\lambda\right), \forall i, \ \mathbf{z}_{x_i} \sim p(\mathbf{z}_x)
\end{aligned}
\tag{35}
$$

then obviously we have that

$$
\begin{aligned}
\lim_{\lambda \to 0_+} p(\mathbf{z}_t) &= \lim_{\lambda \to 0_+} \lim_{N \to +\infty} \sum_{i=1}^{N} \frac{1}{\lambda} C'(\mathbf{z}_{x_i}) \exp\left(-\delta(\mathbf{z}_{x_i}, \mathbf{z}_t)/\lambda\right) \\
&= \lim_{\lambda \to 0_+} \lim_{N \to +\infty} \sum_{i=1}^{N} \frac{1}{\lambda} C' \exp\left(-\delta(\mathbf{z}_{x_i}, \mathbf{z}_t)/\lambda\right),
\end{aligned}
\tag{36}
$$

where $C' = \int_{-\infty}^{\infty} \exp\left(-\delta(\mathbf{0}, \mathbf{z}_t)\right) \mathrm{d}\mathbf{z}_t$. It is obvious that (36) can be viewed as a Kernel Density Estimation over samples $\mathbf{z}_{x_i} \sim p(\mathbf{z}_x)$, and obviously $\lim_{\tau \to 0_+} p(\mathbf{z}_t)$ will converge to $p(\mathbf{z}_x)$ (which is uniform distribution) under quite mild condition (for details of the convergence, refer to Jiang (2017)). $\qquad \square$

**Proof sketch** Similar to hypersphere, the proof of Theorem 4.2 can be done by demonstrating that the right-hand side of Eq. (32) is equal to the right-hand side of Eq. (10) on convex bodies. To achieve this, using Lemma 2, we show that $\mathbf{f}_x \circ \mathbf{g}_x = \mathbf{f}_t \circ \mathbf{g}_t$, and they are independent of the modality-specific variables $\mathbf{m}_x$ and $\mathbf{m}_t$, respectively. Finally, by defining $\mathbf{h} = \mathbf{f}_x \circ \mathbf{g}_x = \mathbf{f}_t \circ \mathbf{g}_t$, and using the inference model (33) and (34), we obtain our result.

**Step I** On convex bodies, and define $\mathbf{h}_x = \mathbf{f}_x \circ \mathbf{g}_x$ and $\mathbf{h}_t = \mathbf{f}_t \circ \mathbf{g}_t$. Consider C1 in Lemma 2, e.g., $\mathbf{h}_x(\mathbf{m}_x, \mathbf{z}_x) = \mathbf{h}_t(\mathbf{m}_t, \mathbf{z}_t)$ almost surely, for pair $\big((\mathbf{m}_x, \mathbf{z}_x), (\mathbf{m}_t, \mathbf{z}_t)\big)$. Similar to Step I in Appendix D.1, by differentiating it with respect to $\mathbf{m}_x$ and $\mathbf{m}_t$, respectively, we can conclude that both $\mathbf{h}_x$ and $\mathbf{h}_t$ are independent of the modality-specific variables $\mathbf{m}_x$ and $\mathbf{m}_t$, respectively, *i.e.*, $\mathbf{h}_x(\mathbf{m}_x, \mathbf{z}_x) = \mathbf{h}_x(\mathbf{z}_x)$ and $\mathbf{h}_t(\mathbf{m}_t, \mathbf{z}_t) = \mathbf{h}_t(\mathbf{z}_t)$. Further, since $\mathbf{h}_x(\mathbf{z}_x) = \mathbf{h}_t(\mathbf{z}_t)$ hold, for all real pairs $(\mathbf{z}_x, \mathbf{z}_t)$ sampled from the conditional distribution $p(\mathbf{z}_t | \mathbf{z}_x)$ defined in Eq. (7), this expression also holds true for $\mathbf{z}_t = \mathbf{z}_x$, which implies $\mathbf{h}_x(\mathbf{z}_x) = \mathbf{h}_t(\mathbf{z}_x)$. As a result, we can obtain: $\mathbf{h}_x = \mathbf{h}_t$.

**Step II** According to the results above: $\mathbf{h}_x(\mathbf{m}_x, \mathbf{z}_x) = \mathbf{h}_x(\mathbf{z}_x)$, $\mathbf{h}_t(\mathbf{m}_t, \mathbf{z}_t) = \mathbf{h}_t(\mathbf{z}_t)$, and $\mathbf{h}_x = \mathbf{h}_t$, by defining $\mathbf{h} \overset{\text{def}}{=} \mathbf{f}_x \circ \mathbf{g}_x = \mathbf{f}_t \circ \mathbf{g}_t$ , we can rewrite Eq. (10) as:

$$
2 \mathop{\mathbb{E}}_{(\mathbf{z}_x, \mathbf{z}_t) \sim p(\mathbf{z}_x, \mathbf{z}_t)} \left[ d\big(\mathbf{h}(\mathbf{z}_x), \mathbf{h}(\mathbf{z}_t)\big)/\tau \right] + \mathop{\mathbb{E}}_{\mathbf{z}_x \sim p(\mathbf{z}_x)} \left[ \log \mathop{\mathbb{E}}_{\mathbf{z}_t \sim p(\mathbf{z}_t)} \left[ e^{-d\big(\mathbf{h}(\mathbf{z}_x), \mathbf{h}(\mathbf{z}_t)\big)/\tau} \right] \right]
$$
$$
+ \mathop{\mathbb{E}}_{\mathbf{z}_t \sim p(\mathbf{z}_t)} \left[ \log \mathop{\mathbb{E}}_{\mathbf{z}_x \sim p(\mathbf{z}_x)} \left[ e^{-d\big(\mathbf{h}(\mathbf{z}_x), \mathbf{h}(\mathbf{z}_t)\big)/\tau} \right] \right]. \tag{37}
$$

We then connect the right-hand side of Eq. (32) with Eq. (37). To this end, since the two terms in the right-hand side of Eq. (32) are symmetrical, we focus on one of the two terms for convenience, *e.g.*, $\mathop{\mathbb{E}}_{\mathbf{z}_x \sim p(\mathbf{z}_x)} \big[ H(p(\mathbf{z}_t | \mathbf{z}_x)), q_{\mathbf{h}}(\mathbf{z}_t | \mathbf{z}_x)) \big]$. It can be shown that:

$$
\mathop{\mathbb{E}}_{\mathbf{z}_x \sim p(\mathbf{z}_x)} \big[ H(p(\mathbf{z}_t | \mathbf{z}_x)), q_{\mathbf{h}}(\mathbf{z}_t | \mathbf{z}_x)) \big] \tag{38}
$$

$$
= \mathop{\mathbb{E}}_{\mathbf{z}_x \sim p(\mathbf{z}_x)} \Big[ \mathop{\mathbb{E}}_{\mathbf{z}_t \sim p(\mathbf{z}_t | \mathbf{z}_x)} \big[ -\log q_{\mathbf{h}}(\mathbf{z}_t | \mathbf{z}_x) \big] \Big] \tag{39}
$$

$$
= \mathop{\mathbb{E}}_{(\mathbf{z}_x, \mathbf{z}_t) \sim p(\mathbf{z}_x, \mathbf{z}_t)} \big[ \delta(\mathbf{h}(\mathbf{z}_x), \mathbf{h}(\mathbf{z}_t))/\tau + \log C_q(\mathbf{z}_x) \big] \tag{40}
$$

$$
= \mathop{\mathbb{E}}_{(\mathbf{z}_x, \mathbf{z}_t) \sim p(\mathbf{z}_x, \mathbf{z}_t)} \big[ \delta(\mathbf{h}(\mathbf{z}_x), \mathbf{h}(\mathbf{z}_t))/\tau \big] + \mathop{\mathbb{E}}_{(\mathbf{z}_x) \sim p(\mathbf{z}_x)} \big[ \log C_q(\mathbf{z}_x) \big] \tag{41}
$$

$$
= \mathop{\mathbb{E}}_{(\mathbf{z}_x, \mathbf{z}_t) \sim p(\mathbf{z}_x, \mathbf{z}_t)} \big[ \delta(\mathbf{h}(\mathbf{z}_x), \mathbf{h}(\mathbf{z}_t))/\tau \big] + \mathop{\mathbb{E}}_{(\mathbf{z}_x) \sim p(\mathbf{z}_x)} \big[ \log \int e^{(-\delta(\mathbf{h}(\mathbf{z}_x), \mathbf{h}(\mathbf{z}_t))/\tau)} \mathrm{d}\mathbf{z}_x \big] \tag{42}
$$

Since $p(\mathbf{z}_x) = |\mathcal{Z}|^{-1}$, and $p(\mathbf{z}_t) = |\mathcal{Z}|^{-1}$ by Lemma 3, Eq. (42) is equal to:

$$= \mathop{\mathbb{E}}_{(\mathbf{z}_x, \mathbf{z}_t) \sim p(\mathbf{z}_x, \mathbf{z}_t)} \left[ \delta(\mathbf{h}(\mathbf{z}_x), \mathbf{h}(\mathbf{z}_t))/\tau \right] + \mathop{\mathbb{E}}_{\mathbf{z}_x \sim p(\mathbf{z}_x)} \left[ \log \mathop{\mathbb{E}}_{\mathbf{z}_t \sim p(\mathbf{z}_t)} \left[ e^{-\delta(\mathbf{h}(\mathbf{z}_x), \mathbf{h}(\mathbf{z}_t))/\tau} \right] \right] + \log|\mathcal{Z}_c| \tag{43}$$

Similarly, for the second term in the right-hand side of Eq. (32), we can proof that:

$$\mathop{\mathbb{E}}_{(\mathbf{z}_t) \sim p(\mathbf{z}_t)} \left[ H(p(\mathbf{z}_x|\mathbf{z}_t)), q_{\mathbf{h}}(\mathbf{z}_x|\mathbf{z}_t)) \right] = \mathop{\mathbb{E}}_{(\mathbf{z}_x, \mathbf{z}_t) \sim p(\mathbf{z}_x, \mathbf{z}_t)} \left[ \delta(\mathbf{h}(\mathbf{z}_x), \mathbf{h}(\mathbf{z}_t))/\tau \right] \tag{44}$$

$$+ \mathop{\mathbb{E}}_{\mathbf{z}_t \sim p(\mathbf{z}_t)} \left[ \log \mathop{\mathbb{E}}_{\mathbf{z}_x \sim p(\mathbf{z}_x)} \left[ e^{-\delta(\mathbf{h}(\mathbf{z}_x), \mathbf{h}(\mathbf{z}_t))/\tau} \right] \right] + \log|\mathcal{Z}_c|. \tag{45}$$

By combining Eq. (43) and Eq. (45), we can conclude the proof.

### E.2 IDENTIFIABILITY RESULT ON CONVEX BODIES

Theorem 4.2 represents a symmetrical adaptation of Theorem 3 from (Zimmermann et al., 2021). This alignment allows us to employ Propositions 4, Lemma 1 and Lemma A from (Zimmermann et al., 2021) to demonstrate that the global minimization of the objective outlined in Eq. (32), as specified in Theorem 4.2, identifies the latent variables $\mathbf{z}_x$, as well as $\mathbf{z}_x$, up to linear transformations. For completeness, a brief proof is provided herein, with comprehensive details available in the original work. Clearly, the global minimum of the cross-entropy between two distributions is reached if they match by value and have the same support. Therefore, for the optimal solution of the objective loss Eq. (10) in Theorem 4.2, we have:

$$p(\mathbf{z}_t|\mathbf{z}_x) = q_{\mathbf{h}}(\mathbf{z}_t|\mathbf{z}_x), \tag{46}$$

This expression also holds true for $\mathbf{z}_t = \mathbf{z}_x$, we have:

$$C_p(\mathbf{z}_x)^{-1} e^{-\delta(\mathbf{z}_x, \mathbf{z}_x)/\lambda} = C_q(\mathbf{z}_x)^{-1} e^{-\delta(\mathbf{h}(\mathbf{z}_x), \mathbf{h}(\mathbf{z}_x))/\tau},$$
$$\Leftrightarrow C_p(\mathbf{z}_x) = C_q(\mathbf{z}_x) \tag{47}$$

As the normalization constants are identical we get for all $\mathbf{z}_x, \mathbf{z}_t$,

$$\delta(\mathbf{z}_x, \mathbf{z}_t) = \lambda\delta(\mathbf{h}(\mathbf{z}_x), \mathbf{h}(\mathbf{z}_t))/\tau. \tag{48}$$

Then, by limiting $\delta$ be an $L^\alpha$ metric for $\alpha \geq 1$, $\alpha \neq 2$ or the $\alpha$-th power of such an $L^\alpha$ metric, using the Theorems 5 and 6 in Zimmermann et al. (2021), $\mathbf{h}$ is a composition of input independent permutations, sign flips and rescaling. In other words, Theorem 4.2 establishes that the latent variables $\mathbf{z}_x$ (and $\mathbf{z}_t$) are identifiable up to a permutation transformation, *i.e.*, the recovered latent variable $\mathbf{f}_x(\mathbf{x})$, obtained through the minimization of Eq. (8), is related to the true $\mathbf{z}_x$ as follows: $\mathbf{f}_x(\mathbf{x}) = \mathbf{P}\mathbf{z}_x + \mathbf{c}$, where $\mathbf{P}$ represents an permutation matrix with scaling, and $\mathbf{c}$ is a constant vector.

## F  DIFFERENCES FROM PREVIOUS ANALYSIS FOR MUTILMODAL CONTRASTIVE LEARNING

This work differs from previous works focusing on identifiability analysis for multimodal settings (Daunhawer et al., 2023; Yao et al., 2023; Gresele et al., 2020) across the following dimensions.

**Modeling Setting**   This work proposes modeling transferable knowledge across modalities by latent coupled variables. In contrast, previous works (Daunhawer et al., 2023; Yao et al., 2023; Gresele et al., 2020) often achieve this by introducing the same/shared variables. The advantages of employing latent coupled variables are thoroughly justified in Section 2. Loosely speaking, From the perspective of model flexibility, the proposed model can be considered a generalization of previous works. This generalization is apparent as the proposed model seamlessly reduces to a single-modal setting when the mixing functions from latent space to observed space are enforced to be identical, and specific variables are omitted.

**Identifiability Results**   The identifiability results obtained in this work diverge from those found in previous works (Daunhawer et al., 2023; Yao et al., 2023), both in terms of breadth and depth of identifiability, due to the introduction of the undirected edge between $\mathbf{z}_x$ and $\mathbf{z}_t$. a) *Breadth of Identifiability:* Unlike earlier works that often achieve only partial identifiability of latent coupled variables $\mathbf{z}_x$ or $\mathbf{z}_t$, *e.g.*, latent content variables but not latent style variables (Daunhawer et al., 2023; Yao et al., 2023), our model extends this scope to ensure complete identifiability of latent coupled variables $\mathbf{z}_x$ and $\mathbf{z}_t$. b) *Depth of Identifiability:* In terms of depth, this work identifies latent coupled variables $\mathbf{z}_x$ and $\mathbf{z}_t$ up to linear or permutation transformations. As a result, after applying a linear ICA method, we can obtain component-wise identifiability, i.e., recovering independent latent variables up to permutation and scaling. This level of precision offers an enhancement over the block identifiability result in previous studies (Daunhawer et al., 2023; Yao et al., 2023), which only identifying latent variables up to a nonlinear invertible mapping, even for independent latent variables. *The differences above in both breadth and depth of identifiability results enable us, for the first time, to unveil the component-wise disentanglement capabilities of multimodal contrastive representation learning.*

**Practical Significance in Real Applications**   Prior studies (Daunhawer et al., 2023; Yao et al., 2023; Gresele et al., 2020) have predominantly relied on simulation experiments, which often encounter a substantial gap between the assumptions made in theoretical analyses and the practical conditions of real-world applications. In contrast, our work bridges this gap by validating our theoretical findings using pre-trained CLIP models on over 16 diverse real-world datasets. This empirical approach not only substantiates the practical effectiveness of our theoretical results but also demonstrates their applicability and robustness in real-world multimodal settings, highlighting a significant departure from previous work in terms of real-world applicability.

## G  DIFFERENCES FROM PREVIOUS ANALYSES FOR SINGLE-MODAL CONTRASTIVE LEARNING

This work sets itself apart from prior studies focused on the analysis of single-modal contrastive learning (Zimmermann et al., 2021; Von Kügelgen et al., 2021) in the following key aspects.

**Problem Context**   Previous works primarily address single-modal scenarios, whereas our proposed model extends this framework to the more complex multimodal domain. This extension can be viewed as a generalization of prior approaches. Specifically, our model naturally reduces to a single-modal setting when the mixing functions from the latent space to the observed space are identical, and certain variables are omitted. By expanding the scope to multimodal data, our approach addresses the limitations of prior studies and provides a more comprehensive understanding of contrastive learning.

**Technical Perspective**   Addressing multimodal settings requires significantly broader technical developments compared to single-modal analyses. To this end, we developed Theorem 3.1, which generalizes the asymptotic analysis of contrastive learning to the multimodal context, providing a robust theoretical foundation. Bridging the gap between single-modal and multimodal settings

also necessitated novel theoretical insights. For instance, Theorem 4.1 and Theorem 4.2 establish critical connections between multimodal contrastive learning and traditional single-modal frameworks, enabling a unified understanding across these domains. These results not only expand the applicability of contrastive learning but also highlight the intricate dependencies introduced by multimodal data.

**New Insights** In the multimodal context, a key challenge is effectively modeling the connections between different modalities. This motivates the central insight of our work: latent coupled variables, linked by a unidirectional edge, provide a foundation for exploring whether partial causal models are sufficient for multimodal learning. As highlighted in the introduction, we offer theoretical support for the success of multimodal contrastive learning, including guarantees for its disentanglement capabilities. From a practical perspective, we recommend refining representations from pre-trained CLIP-like models rather than using them directly. Specifically, applying linear ICA methods, such as FastICA (aligned with assumptions on the hypersphere), or combining PCA and FastICA (aligned with assumptions on convex bodies), can enhance performance on tasks that rely on disentangled representations. These insights not only validate the robustness of our theoretical findings but also emphasize their practical significance in real-world applications.

## H  MORE RESULTS ON CELEBA

Figures 5 - 7 illustrate the 16 distinct disentangled representations obtained using pre-trained CLIP with FastICA. Interestingly, our method achieves competitive results compared to specialized disentanglement techniques, such as FactorVAE (Kim & Mnih, 2018) and $\beta$-TCVAE (Chen et al., 2018). Specifically, FactorVAE identified 8 disentangled attributes, while $\beta$-TCVAE reported 15, whereas our approach successfully discerns 16 distinct disentangled representations.

It is important to note that this comparison is not meant to position our method as a more effective disentanglement technique. Rather, our experiments are designed solely to validate our theoretical findings. We present this comparison to provide insight into the potential of leveraging CLIP for learning disentangled representations, thereby motivating future research in this direction. A particularly interesting avenue could be exploring how disentanglement capabilities relate to the manipulation of pre-trained vision models, such as diffusion models.

## I  MORE RESULTS ON IMAGENET-TYPE DATA

Table 3: Quantitative results for 16-shot transfer learning and domain generalization by different methods. Lin. P. (Linear Probe).

| ENCODERS | METHODS | SOURCE | TARGET (IMAGENET-) | | | | |
|---|---|---|---|---|---|---|---|
| | | IMAGENET | V2 | SKETCH | R | A | AVG. |
| RN50 | LIN. P. | 55.36 | 45.45 | 18.22 | 34.09 | 12.52 | 27.77 |
| | LIN. P. W/ FASTICA | 57.82 | 47.78 | 19.77 | 38.05 | 13.15 | 29.69 |
| | LIN. P. W/ PCA AND FASTICA | 57.37 | 47.67 | 20.39 | 38.76 | 12.89 | 29.93 |
| RN101 | LIN. P. | 60.98 | 50.36 | 25.80 | 46.61 | 18.64 | 35.35 |
| | LIN. P. W/ FASTICA | 61.86 | 51.85 | 27.29 | 49.29 | 19.89 | 37.08 |
| | LIN. P. W/ PCA AND FASTICA | 61.58 | 51.44 | 28.86 | 50.32 | 19.97 | 37.64 |
| VIT32 | LIN. P. | 60.76 | 50.92 | 28.81 | 49.18 | 19.72 | 37.15 |
| | LIN. P. W/ FASTICA | 61.94 | 51.95 | 30.30 | 51.82 | 20.81 | 38.72 |
| | LIN. P. W/ PCA AND FASTICA | 62.00 | 52.39 | 30.39 | 51.61 | 20.96 | 38.84 |
| VIT16 | LIN. P. | 67.17 | 57.01 | 35.43 | 60.96 | 35.41 | 47.20 |
| | LIN. P. W/ PCA AND FASTICA | 68.12 | 58.45 | 38.41 | 63.89 | 37.17 | 49.48 |
| | LIN. P. W/ FASTICA | 67.96 | 58.38 | 38.75 | 65.45 | 38.28 | 50.22 |

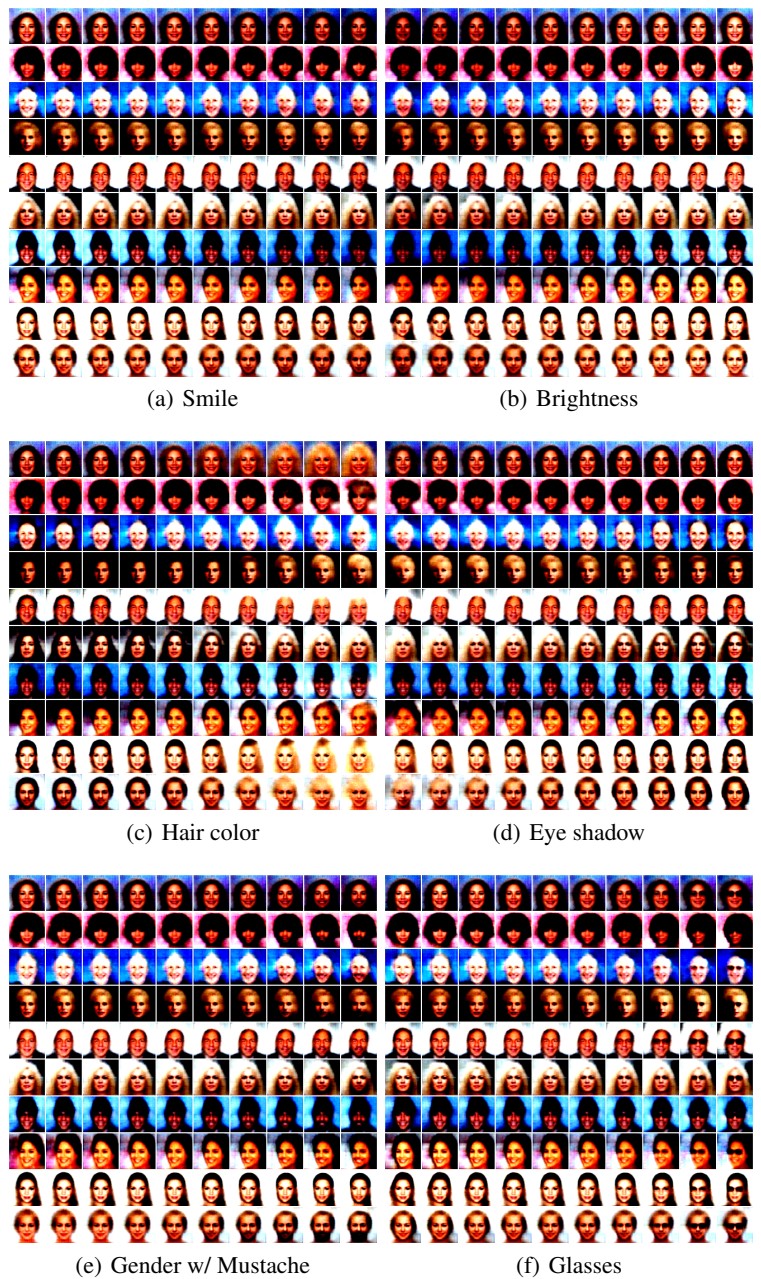

(a) Smile  (b) Brightness

(c) Hair color  (d) Eye shadow

(e) Gender w/ Mustache  (f) Glasses

Figure 5: Disentangled Representations learned by combining pre-train CLIP and FastICA.

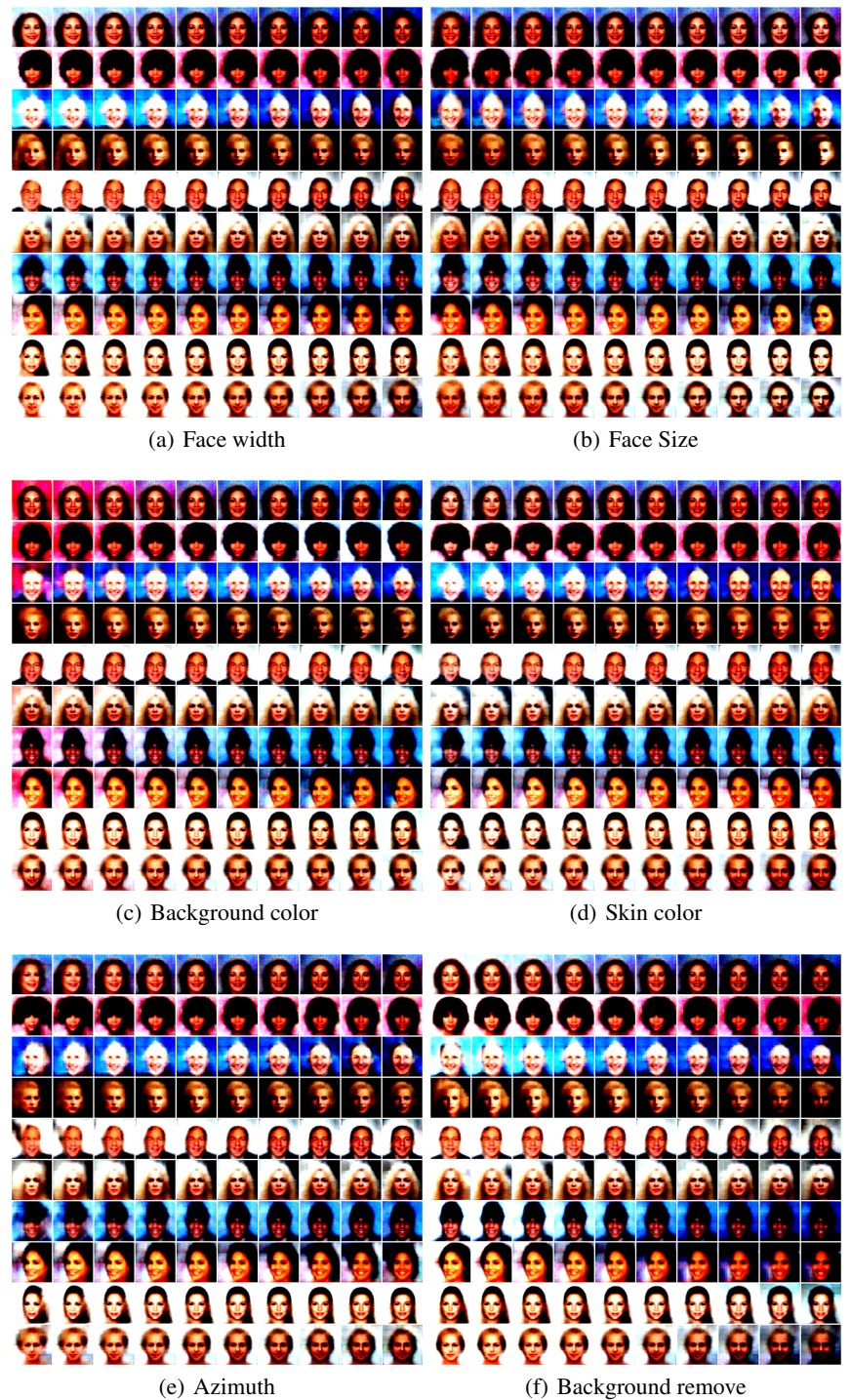

(a) Face width

(b) Face Size

(c) Background color

(d) Skin color

(e) Azimuth

(f) Background remove

Figure 6: Disentangled Representations learned by combining pre-train CLIP and FastICA.

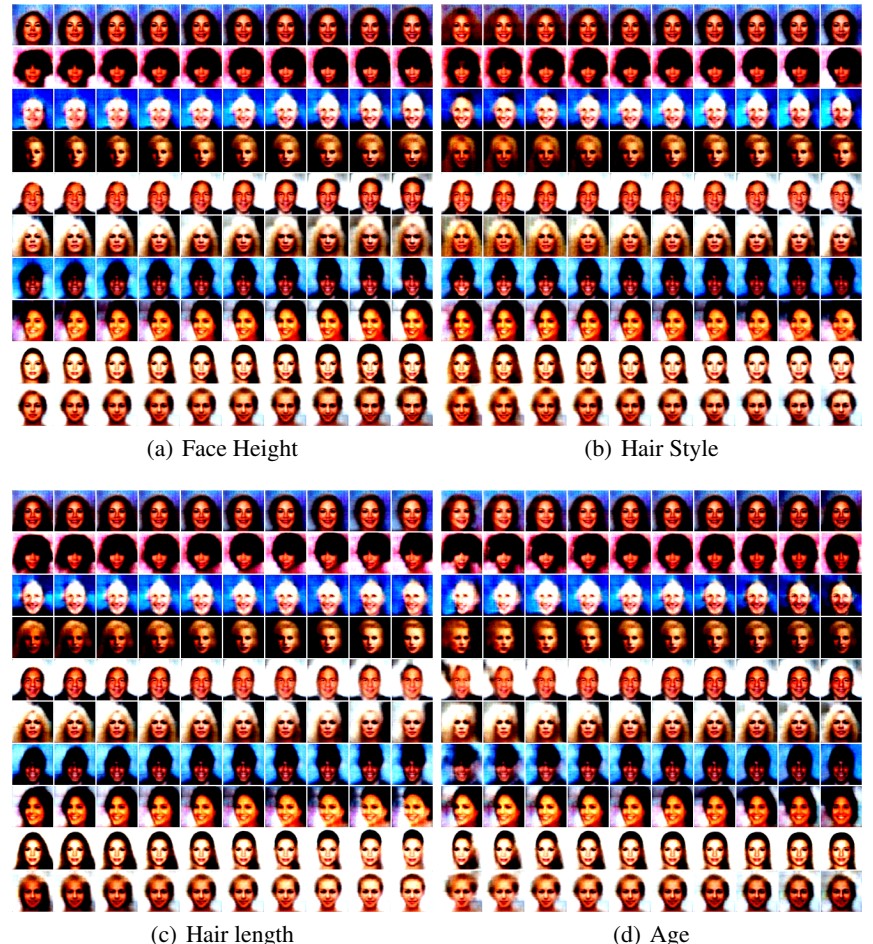

(a) Face Height

(b) Hair Style

(c) Hair length

(d) Age

Figure 7: Disentangled Representations learned by combining pre-train CLIP and FastICA.

Table 4: Quantitative results for 8-shot transfer learning and domain generalization by different methods. Lin. P. (Linear Probe).

| ENCODERS | METHODS | SOURCE | TARGET (IMAGENET-) | | | | |
|---|---|---|---|---|---|---|---|
| | | IMAGENET | V2 | SKETCH | R | A | AVG. |
| RN50 | LIN. P. | 49.33 | 40.83 | 15.06 | 31.23 | 10.99 | 24.53 |
| | LIN. P. W/ FASTICA | 51.99 | 43.58 | 15.47 | 34.35 | 12.85 | 26.56 |
| | LIN. P. W/ PCA AND FASTICA | 51.42 | 42.93 | 17.28 | 35.53 | 12.33 | 27.02 |
| RN101 | LIN. P. | 55.41 | 46.04 | 23.38 | 43.26 | 16.88 | 32.39 |
| | LIN. P. W/ FASTICA | 56.59 | 47.47 | 22.09 | 44.59 | 18.39 | 33.14 |
| | LIN. P. W/ PCA AND FASTICA | 55.84 | 46.59 | 23.68 | 44.94 | 18.25 | 33.37 |
| VIT32 | LIN. P. | 55.17 | 46.11 | 25.53 | 45.32 | 18.35 | 33.83 |
| | LIN. P. W/ FASTICA | 56.90 | 47.96 | 27.62 | 49.13 | 20.31 | 36.26 |
| | LIN. P. W/ PCA AND FASTICA | 55.83 | 46.55 | 26.54 | 46.77 | 18.80 | 34.67 |
| VIT16 | LIN. P. | 61.82 | 52.34 | 32.26 | 55.93 | 32.63 | 43.29 |
| | LIN. P. W/ FASTICA | 63.55 | 54.81 | 34.21 | 61.54 | 38.21 | 47.29 |
| | LIN. P. W/PCA AND FASTICA | 63.47 | 54.32 | 35.83 | 61.88 | 37.35 | 47.36 |

Table 5: Quantitative results for 4-shot transfer learning and domain generalization by different methods. Lin. P. (Linear Probe).

| ENCODERS | METHODS | SOURCE | TARGET (IMAGENET-) | | | | |
| --- | --- | --- | --- | --- | --- | --- | --- |
| | | IMAGENET | V2 | SKETCH | R | A | AVG. |
| RN50 | LIN. P. | 41.34 | 33.67 | 11.55 | 26.27 | 9.67 | 20.29 |
| | LIN. P. W/ FASTICA | 44.10 | 36.07 | 12.75 | 30.15 | 11.64 | 22.65 |
| | LIN. P. W/ PCA AND FASTICA | 42.86 | 35.38 | 12.29 | 28.81 | 9.79 | 21.57 |
| RN101 | LIN. P. | 48.23 | 39.53 | 18.80 | 38.10 | 14.32 | 27.69 |
| | LIN. P. W/ FASTICA | 49.43 | 41.02 | 17.49 | 39.33 | 15.25 | 28.27 |
| | LIN. P. W/ PCA AND FASTICA | 49.01 | 40.25 | 19.26 | 39.71 | 14.75 | 28.49 |
| VIT32 | LIN. P. | 47.82 | 39.53 | 21.51 | 40.94 | 15.99 | 29.49 |
| | LIN. P. W/ FASTICA | 49.43 | 40.66 | 22.66 | 41.78 | 16.41 | 30.38 |
| | LIN. P. W/ PCA AND FASTICA | 49.48 | 41.09 | 23.72 | 43.48 | 16.77 | 31.27 |
| VIT16 | LIN. P. | 54.30 | 46.06 | 27.58 | 50.76 | 29.24 | 38.41 |
| | LIN. P. W/ FASTICA | 56.65 | 48.18 | 28.27 | 55.50 | 33.39 | 41.33 |
| | LIN. P. W/ PCA AND FASTICA | 56.16 | 47.46 | 30.21 | 55.49 | 31.71 | 41.22 |

Table 6: Quantitative results for 1-shot transfer learning and domain generalization by different methods. Lin. P. (Linear Probe).

| ENCODERS | METHODS | SOURCE | TARGET (IMAGENET-) | | | | |
| --- | --- | --- | --- | --- | --- | --- | --- |
| | | IMAGENET | V2 | SKETCH | R | A | AVG. |
| RN50 | LIN. P. | 21.74 | 18.24 | 5.68 | 15.41 | 6.55 | 11.47 |
| | LIN. P. W/ FASTICA | 23.22 | 19.68 | 6.37 | 13.84 | 7.21 | 11.77 |
| | LIN. P. W/ FASTICA | 24.06 | 20.26 | 6.85 | 17.54 | 8.05 | 13.18 |
| RN101 | LIN. P. | 26.05 | 21.48 | 9.90 | 23.85 | 10.17 | 16.35 |
| | LIN. P. W/ FASTICA | 27.50 | 23.33 | 8.35 | 17.87 | 10.71 | 15.07 |
| | LIN. P. W/ PCA AND FASTICA | 28.50 | 24.17 | 11.63 | 26.38 | 12.28 | 18.62 |
| VIT32 | LIN. P. | 26.99 | 22.99 | 11.93 | 25.25 | 11.56 | 17.93 |
| | LIN. P. W/ FASTICA | 29.21 | 24.80 | 9.97 | 21.23 | 12.23 | 17.06 |
| | LIN. P. W/ PCA AND FASTICA | 29.05 | 24.45 | 12.39 | 27.61 | 12.56 | 19.25 |
| VIT16 | LIN. P. | 32.42 | 27.64 | 16.34 | 34.28 | 21.84 | 25.02 |
| | LIN. P. W/ FASTICA | 34.35 | 29.31 | 13.91 | 28.61 | 23.24 | 23.77 |
| | LIN. P. W/ PCA AND FASTICA | 35.20 | 30.26 | 19.17 | 38.87 | 26.41 | 28.68 |

# J    MORE RESULTS ON FEW-SHOT LEARNING TASK

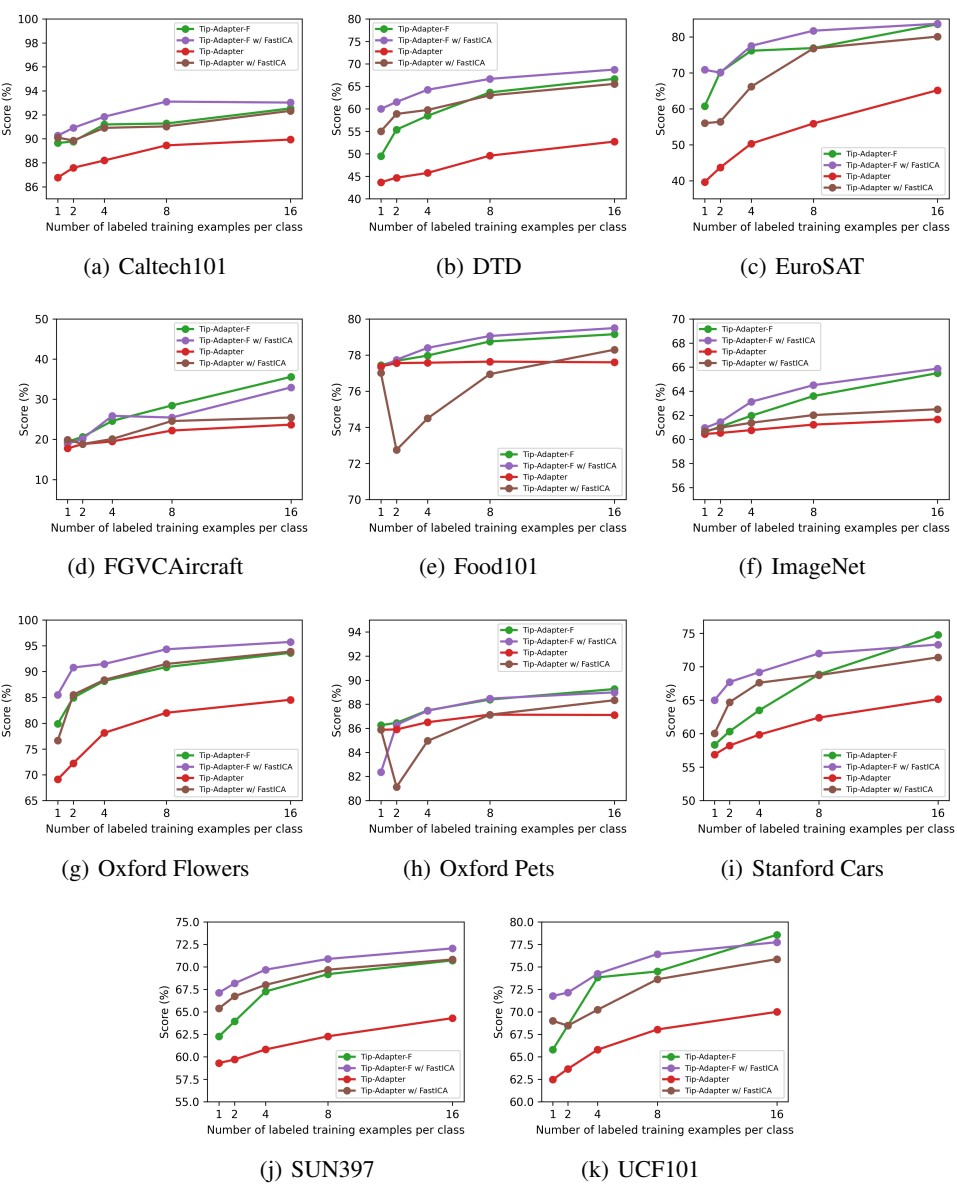

Figure 8: More results on few-shot learning task: A comparison of top-1 accuracy (%) achieved by various few-shot CLIP adaptation methods across 11 datasets, including ImageNet (Deng et al., 2009), Caltech101 (Fei-Fei et al., 2004), FGVCAircraft (Maji et al., 2013), UCF101 (Soomro et al., 2012), EuroSAT (Helber et al., 2019), Flowers102 (Nilsback & Zisserman, 2008), StanfordCars (Krause et al., 2013), DTD (Cimpoi et al., 2014), Food101 (Bossard et al., 2014), OxfordPets (Parkhi et al., 2012), and, SUN397 (Xiao et al., 2010). The x-axis indicates the number of training examples per class.The incorporation of FastICA notably enhances the performance of the original methods, Tip-Adapter and Tip-Adapter-F, proposed by (Zhang et al., 2022a).

## K   IMPLEMENTATION DETAILS

We perform all experiments using the GPU RTX 4090, equipped with 32 GB of memory.

**Synthetic Data**   We consider latent coupled variables $\mathbf{z}_x$ and $\mathbf{z}_t$, each with a dimensionality of 10. Additionally, we have modality-specific latent variables $\mathbf{m}_x$ and $\mathbf{m}_t$, both set to a dimension of 5. The process begins with sampling from the marginal distribution $p(\mathbf{z}_x)$, and the samples of modality-specific latent variables $\mathbf{m}_x$ and $\mathbf{m}_t$ are obtained by sampling from Gaussian distributions with zero mean and one variance. We then create real pairs by sampling from the conditional distribution $p(\mathbf{z}_t|\mathbf{z}_x)$. The observational data $\mathbf{x}$ and $\mathbf{t}$ are generated using two different Multi-Layer Perceptrons (MLPs). Specifically, we utilize MLPs comprising three hidden layers with leaky ReLU units and random weights. To ensure the invertibility of the MLP g, we carefully control the condition number of the weight matrices. For our encoders concerning both $\mathbf{z}_t$ and $\mathbf{z}_x$, we adopt an MLP architecture with leaky ReLU units.

**Evaluation:**   To evaluate the linear identifiability result established in Corollary 1, we assess how well the learned representations $\hat{\mathbf{z}}_x$ preserve the structure of the ground-truth latent variables $\mathbf{z}_x$ up to a linear transformation. Specifically, we perform the following steps:

1. **Learned Representations Extraction:** We first obtain representations $\hat{\mathbf{z}}_x$ learned by multimodal contrastive learning.

2. **Linear Regression Fitting:** We fit a linear regression model of the form:

$$\hat{\mathbf{z}}_x = \hat{\mathbf{A}}\mathbf{z}_x + \hat{\mathbf{c}} + \epsilon,$$

   where $\hat{\mathbf{A}}$ is a learned transformation matrix, $\hat{\mathbf{c}}$ is an offset vector, and $\epsilon$ represents residual errors.

3. **Coefficient of Determination ($R^2$) Computation:** We compute the $R^2$ score, defined as:

$$R^2 = 1 - \frac{\sum_i \|\hat{\mathbf{z}}_{x,i} - (\hat{\mathbf{A}}\mathbf{z}_{x,i} + \hat{\mathbf{c}})\|^2}{\sum_i \|\hat{\mathbf{z}}_{x,i} - \bar{\hat{\mathbf{z}}}_x\|^2},$$

   where $\bar{\hat{\mathbf{z}}}_x$ is the mean of $\hat{\mathbf{z}}_x$. This metric quantifies how well the learned representations can be linearly mapped to the true latent variables.

4. **Analysis Under Different Assumption Violations:** We repeat the evaluation under settings that both satisfy and violate the theoretical assumptions, as listed in Table 1, allowing us to empirically assess the robustness of the identifiability results.

By reporting the $R^2$ scores across different conditions, we quantify the extent to which multimodal contrastive learning successfully recovers the latent variables up to a linear transformation.

to evaluate permutation identifiability result in Corollary 2, we employ the mean correlation coefficient (MCC) between the ground-truth $\mathbf{z}_x$ and representations $\mathbf{f}_x(\mathbf{x})$ learned by multimodal contrastive learning. To compute MCC, we follow these steps:

1. **Compute Correlation Coefficients:** We first calculate the correlation coefficients between all pairs of ground-truth source variables and representations learned by multimodal contrastive learning. Specifically, for each pair of source component $\mathbf{z}_{x,i}$ and recovered latent component $\hat{\mathbf{z}}_{x,j}$, we compute the Pearson correlation coefficient:

$$\rho_{i,j} = \frac{\mathrm{Cov}(\mathbf{z}_{x,i}, \hat{\mathbf{z}}_{x,j})}{\sigma_{\mathbf{z}_{x,i}}\sigma_{\hat{\mathbf{z}}_{x,j}}}, \tag{49}$$

   where $\mathrm{Cov}(\cdot,\cdot)$ denotes the covariance, and $\sigma_{\mathbf{z}_{x,i}}$ and $\sigma_{\hat{\mathbf{z}}_{x,j}}$ are the standard deviations of the respective components.

2. **Solve the Linear Sum Assignment Problem:** Since the estimated components may be permuted relative to the ground-truth variables, we solve a linear sum assignment problem to determine the optimal one-to-one mapping between the ground-truth and the learned representations. The goal is to maximize the total absolute correlation across all assigned pairs.

| |
|---|
| ReLU(BN(ConvTranspose2d(512, 512, kernelsize=1, stride=1, padding=0))) |
| ReLU(BN(ConvTranspose2d(512, 64, kernelsize=4, stride=1, padding=0))) |
| ReLU(BN(ConvTranspose2d(64, 64, kernelsize=4, stride=1, padding=0))) |
| ReLU(BN(ConvTranspose2d(64, 32, kernelsize=4, stride=1, padding=0))) |
| ReLU(BN(ConvTranspose2d(32, 32, kernelsize=4, stride=1, padding=0))) |
| ConvTranspose2d(32, 3, kernelsize=4, stride=2, padding=1) |

Table 7: Decoder for the image data.

3. **Compute the Mean Correlation Coefficient (MCC):** Given the optimal assignment of the ground-truth variables to the learned representations, we compute the mean of the absolute values of the assigned correlation coefficients:

$$\text{MCC} = \frac{1}{d} \sum_{i=1}^{d} |\rho_{i,\pi(i)}|, \tag{50}$$

where $\pi(i)$ denotes the index of the assigned representation corresponding to the $i$th latent variable, and $d$ is the number of latent variables.

A high MCC indicates that the learned representation closely match the true source variables, up to permutation transformations, thereby validating the identifiability of the learned representations.

**Disentangled Representation Learning on CelebA**  To obtain disentangled representations for the CelebA dataset, we initially employ the FastICA implementation available in the scikit-learn software on the representations extracted from the pretrained ViT-B/32 encoder. Subsequently, we train the decoder, as outlined in Table 7, utilizing Mean Squared Error (MSE) loss.

**Experiments of Linear Probe**  In our experiments with ImageNet-Type data, we utilized the PCA and FastICA implementations provided by scikit-learn. For our proposed method, which combines PCA and ICA, we configured the number of components to 500 for PCA, and for FastICA, we set it to 160 for 1, 2, and 4-shot learning scenarios, and 200 for 8 and 16-shot learning scenarios. When employing ICA alone, we chose to use 300 components. For the proposed method with ICA only, we set number of components to 300. Following the setting of linear probe in CLIP, we train a logistic regression classifier using scikit-learn's L-BFGS implementation, with maximum 1,000 iterations. We determine the L2 regularization strength using a hyperparameter sweep on the validation sets over the range between $10^{-6}$ and $10^{6}$ , with 96 logarithmically spaced steps. To save compute required for the sweeps, we perform a parametric binary search and iteratively halves the interval around the peak until it reaches a resolution of 8 steps per decade. The hyperparameter sweeps are performed on a validation split of each dataset.

**FastICA as a plug-and-play Tool.**  We incorporate FastICA in the framework proposed in (Zhang et al., 2022a) to enhance its ability for few shot learning. The framework consists of two primary modules: one keeps the zero-shot capabilities of pre-trained CLIP, ensuring effective utilization of prior knowledge, while the other, the cache module, constitutes the central contribution of the work. The cache module endeavors to transfer knowledge from labeled training samples. Given the above, we integrate FastICA into the cache module, preserving the invaluable prior knowledge derived from the zero-shot abilities of pre-trained CLIP. For parameter settings in FastICA, we opted for 100 components for the majority of datasets. Specifically, we assigned 350 components for the ImageNet dataset, 300 components for the OxfordPets dataset, and 50 components for the EuroSAT dataset. A learning rate of 0.1 was employed for implementation. For the remaining parameter settings, we adhered to the specifications outlined by (Zhang et al., 2022a).

## L  DISCUSSIONS ON FASTICA VS. PCA FOLLOWED BY FASTICA

Our theoretical findings are based on two distinct assumptions: one on the hypersphere (Sec. 4.1) and the other on convex bodies (Sec. 4.2). Each of these assumptions motivates a corresponding practical method for real applications, namely FastICA, and PCA followed by FastICA, respectively. In practice, however, the true latent generative process is typically unknown, making it difficult to determine a priori which method is more appropriate. From an empirical standpoint, we observe that face image datasets, such as CelebA, tend to align more closely with the hypersphere assumption. This observation is supported both by our experiments on CelebA, where learning disentangled representations under the hypersphere assumption improves performance, such as dynamic facial expressions generation, dynamic facial expression transfer(Otberdout et al., 2020), face recognition (Zhong et al., 2021; Liu et al., 2017). Moreover, consistent with the main motivation for learning disentangled representations, we find that the representations obtained under the hypersphere assumption lead to improved performance on related downstream tasks, further validating its practical usefulness.

## M  ACKNOWLEDGMENT OF LLMS USAGE

We acknowledge that large language models (LLMs) were used in this work only for word-level tasks, including correcting typos, improving grammar, and refining phrasing. No substantive content, results, or scientific interpretations were generated by LLMs. All scientific ideas, analyses, and conclusions presented in this manuscript are solely the work of the authors.

## N  HIGH-LEVEL DISCUSSION AND RATIONALE FOR THE USED ASSUMPTIONS

Our identifiability analysis, like most theoretical works on latent variable recovery, relies on specific parametric assumptions about the underlying Data Generating Process (DGP) for $\mathbf{z}_x$ and $\mathbf{z}_t$. While the exact DGP of large-scale multimodal data is unknown, these assumptions are essential for theoretical tractability and are motivated by prevalent machine learning practices and geometrical constraints.

### N.1  RATIONALE AND INTERPRETATION OF ASSUMPTIONS

We introduce two sets of assumptions, primarily centered on the nature of the **latent space geometry** and the **distributional modeling** of the coupled variables.

**Hypersphere Assumptions (Eq. 4)**

- **Latent Space Geometry** ($\mathbb{S}^{M-1}$)**:** The assumption that the latent space resides on a Hypersphere is motivated by consistency with models trained via MMCL. Specifically, modern architectures like CLIP typically enforce $L_2$ normalization on their embeddings, which geometrically constrains the learned representations to lie on the unit sphere. Therefore, assuming the underlying generative factors are also on the hypersphere is a natural choice for space matching. Moreover, this geometry is inspired by prior work in Zimmermann et al. (2021), which demonstrates its potential for achieving disentanglement in the single-modal contrastive learning context.
- **Marginal Distribution** $p(\mathbf{z}_x)$ **as Uniform:** This represents a maximum-entropy assumption. Essentially, in the absence of specific prior knowledge, we assume that the distribution of the shared latent variables, $\mathbf{z}_x$, is uniform across the latent space.
- **Conditional Distribution** $p(\mathbf{z}_t|\mathbf{z}_x)$ **as von Mises-Fisher (vMF) Distribution:** The vMF distribution is the natural counterpart of the Gaussian distribution defined on a sphere. Its parameterized form models the semantic coupling by formalizing the objective of MMCL: given a factor $\mathbf{z}_x$, the distribution expects its positive pair $\mathbf{z}_t$ to be concentrated nearby with high probability. The alignment parameter $k\mathbf{z}_t^T\mathbf{z}_x$ precisely quantifies the strength of this shared semantic information across modalities.

**Convex Body Assumptions (Eq. 7)**  The assumptions for the convex body (e.g., hyperrectangle) case provide an alternative geometric setting, often preferred in classic disentanglement works.

- **Latent Space ($\mathcal{Z}_c$) as a Bounded Convex Body:** This definition specifies a non-spherical, bounded space, which is typically crucial for ensuring identifiability and has been used in previous related works Zimmermann et al. (2021).

- **Conditional Distribution $p(\mathbf{z}_t|\mathbf{z}_x)$ as Exponential Distribution:** The mathematical form $e^{-\delta(\mathbf{z}_t, \mathbf{z}_x)/\lambda}$ models the coupling relationship by assuming the likelihood decays exponentially with the distance ($\delta$, a distance metric induced by a norm between the coupled variables. This implies that given $\mathbf{z}_x$, the paired variable $\mathbf{z}_t$ is likely to be found in its immediate vicinity.

## N.2    LIMITATIONS AND PRACTICAL IMPLICATIONS

While these assumptions are theoretically sufficient for identifiability, as we have shown, their strict adherence in real-world scenarios is challenging to verify. This difficulty arises because the true data-generating process is unknown, making direct verification of conditions generally impossible. As is common in practice, performance gains on downstream tasks are therefore used as a surrogate to assess the plausibility of the theoretical assumptions. In particular, if the methods derived from our identifiability theorems (e.g., using FastICA or PCA+FastICA to recover disentangled representations) consistently yield strong improvements across diverse downstream tasks—as demonstrated in our extensive experiments on few-shot learning and domain generalization—then it is reasonable to infer that the underlying assumptions are either satisfied or, more likely, approximated sufficiently well for the theory to be practically meaningful.

## O    A FORMAL DEFINITION OF DISENTANGLEMENT

**Definition 1** (Component-wise Disentanglement). *A representation $\mathbf{f}_x(\mathbf{x})$ (and symmetrically, $\mathbf{f}_t(\mathbf{t})$) learned by MMCL is defined as **Component-wise Disentangled** if two conditions are met:*

1. ***Factor Independence (Prerequisite):** The components of the underlying latent coupled variable $\mathbf{z}_x$ (and $\mathbf{z}_t$) are mutually statistically independent.*

2. ***Identifiability up to Trivial Transformation:** The representation $\mathbf{f}_x(\mathbf{x})$ (and symmetrically, $\mathbf{f}_t(\mathbf{t})$) is related to the true latent variable $\mathbf{z}_x$ (and $\mathbf{z}_t$) through a simple, invertible transformation $\mathbf{T}$ up to a constant $\mathbf{c}$:*

$$\mathbf{f}_x(\mathbf{x}) = \mathbf{T}\mathbf{z}_x + \mathbf{c} \quad \textit{and symmetrically} \quad \mathbf{f}_t(\mathbf{t}) = \mathbf{T}'\mathbf{z}_t + \mathbf{c}'$$

*where $\mathbf{T}$ (and $\mathbf{T}'$) is a matrix of trivial ambiguity, specifically:*

- *$\mathbf{T}$ is an orthogonal matrix (in the hypersphere latent space, Corollary 1).*
- *$\mathbf{T}$ is a permutation matrix with scaling (in the convex body latent space, Corollary 2).*

*This result guarantees that the shared, independent components of $\mathbf{z}_x$ (and $\mathbf{z}_t$) can be uniquely recovered by resolving the ambiguity $\mathbf{T}$ using post-hoc linear methods, such as FastICA.*

## P    QUANTITATIVE VALIDATION ON HIGH-DIMENSIONAL IMAGE

To provide a more direct and quantitative assessment of our theory's disentanglement capabilities in higher-dimensional and more complex settings, we utilize the *Multimodal3DIdent* dataset (Daunhawer et al., 2023), which provides paired image and text samples with complete ground-truth latent factors. We validate the identifiability of the shared latent variables ($\mathbf{z}_x$ and $\mathbf{z}_t$) that jointly influence both the image and text modalities. For shared factors, we consider the object's *shape* (7 discrete values) and its position (`object_xpos`, `object_ypos`, `object_zpos`). The remaining factors are treated as modality-specific; see Daunhawer et al. (2023) for further details. The table below presents the $\mathbf{R}^2$ scores for recovering the ground-truth shared latent factors from both the image ($\mathbf{z}_x$) and text ($\mathbf{z}_t$) factors.

The image branch ($\mathbf{z}_x$) achieves near-perfect recovery ($\mathbf{R}^2 \approx 0.97$), robustly validating our theoretical framework's ability to identify and unmix latent factors from complex, high-dimensional inputs.

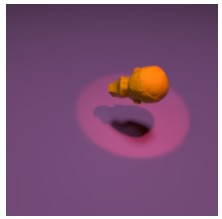

| Modality | Representation | $R^2$ Score (Recovery of $z$) |
|---|---|---|
| Image | $\mathbf{z}_x$ | $0.97 \pm 0.05$ |
| Text | $\mathbf{z}_t$ | $0.75 \pm 0.04$ |

Figure 9: A sample from Multimodal3DIdent (Left). The corresponding text is: 'The top-right of the image shows a "tab:orange" colored head'. Identifiability Scores ($R^2$) (Right).

Recovery performance for the text representation ($\mathbf{z}_t$) is slightly lower ($R^2 \approx 0.75$) (Similar observations were also reported by Daunhawer et al. (2023).), which we attribute primarily to the violation of the idealized continuous assumptions inherent to the text modality—text factors (e.g., color) are often represented as discrete, named values, which conflicts with the continuous assumptions. Overall, these results suggest that our linear identifiability results extend effectively to high-dimensional image data.

