# OpenReview forum: "Beyond DAGs: A Latent Partial Causal Model for Multimodal Learning"
_ICLR.cc/2026/Conference — ICLR 2026 Poster_

### Official Review · Reviewer_LUiv · 2025-10-27

**Soundness:** 3
**Presentation:** 3
**Contribution:** 2
**Rating:** 6
**Confidence:** 3

**Summary:**

The paper argues that a single DAG poorly captures large-scale multimodal generative processes and proposes a latent partial causal model with coupled visual/text factors ($z_x$, $z_t$) linked by undirected cross-modal dependencies. Under distributional and invertibility assumptions, it shows MMCL representations are identifiable up to simple transforms with respect to these latent variables, implying component-wise disentanglement potential. Using pretrained CLIP, few-shot and domain-generalization experiments report consistent gains across several real datasets.

**Strengths:**

- This paper is well-motivated. The author systematically argues that the generation process of large-scale multimodal alignment data has heterogeneity and may have opposite causal directions. The assumption of a single DAG is too limited and needs to be relaxed to describe "partial causality", which is consistent with the actual collection process of web scale text image paired data.

- The writing is overall clear. The contribution and limitations of the theory proposed in the article are analyzed.

**Weaknesses:**

- The identifiability proved here is representation-level (up to simple transforms) rather than causal-effect identifiability.  I suggest the authors explicitly clarify this distinction, since the use of a "causal graph" and the term "partial causal model" may easily lead readers to misinterpret that causal-effect identification is being addressed.
- According to the paper’s theoretical analysis, the identifiability and component-wise disentanglement properties of MMCL under the proposed latent partial causal model should not be specific to the visual branch. Yet the empirical evidence focuses almost exclusively on disentangling CLIP’s image features. If the theory is modality-agnostic, we would expect similar benefits when (i) disentangling the text branch and (ii) optimizing truly multimodal objectives.
- Typos or Mistakes:
  - Line 169, $\tau$ should be a temperature hyper-parameter.
  - In the caption of Table 2, there are two extra ①.
  - Line 1730, Sec. ??.
- No additional method or component is introduced to correspond to the theory; the paper relies on existing techniques (PCA and FastICA).

**Questions:**

See Weaknesses.

---

> ### Author Response · Authors · 2025-11-24
>
> Before responding, we would like to sincerely thank you for taking the time to review our work.
>
> -----
>
> **Q1: The identifiability proved here is representation-level (up to simple transforms) rather than causal-effect identifiability. I suggest the authors explicitly clarify this distinction.}
>
> **R1:** We thank the reviewer for this comment. We fully agree that the distinction between causal discovery and causal effect need to be explicitly clarified, particularly given our use of the term partial causal model.
>
> To address this, we have revised the paper to clearly highlight this distinction by adding a new section (Section O), which provides a formal definition of our disentanglement result (also suggested by Reviewer QDaK). Under this definition, our identifiability results can be rigorously interpreted as guaranteeing the recovery of latent partial causal factors up to simple transformations, rather than causal-effect identification. Thanks again.
>
>
> **Q2: We would expect similar benefits when (i) disentangling the text branch and (ii) optimizing truly multimodal objectives.**
>
> **R2:** We thank the reviewer for this valuable suggestion.
>
> *Evaluate Multimodal Objective*. We clarify that the experiments in the Linear Probe and Tip-Adapter tasks already tests the quality of a multimodal objective:
>
> *Linear Probe Evaluates Multimodal Transferability:* The Linear Probe task, as applied to CLIP, is inherently multimodal because the image features ($\mathbf{f}_x$) are extracted from an encoder trained with the MMCL. The feature quality is therefore inextricably tied to the success of the cross-modal alignment objective. Our consistent performance improvement on Linear Probe (Few-Shot and Domain Generalization, e.g., Table 2) confirms that disentangling these multimodally-shaped features boosts their semantic purity and transferability.
>
> *Tip-Adapter Relies on Zero-Shot Branch:* The Tip-Adapter framework directly exploits the principle of cross-modal alignment by combining image features with text-encoded knowledge, by Zero-Shot classifier branch of CLIP.
>
> *New Experiment on the Text Branch Only* To further prove that our theory disentangling text only, we introduce the following experiment: We augment the Tip-Adapter framework with a Text-Alignment Module for few-shot learning, similar to Figure 4. For each query image, we use an external model (LLaVA) to generate a caption, which is then encoded by CLIP. We then apply ICA to both the query text feature ($\mathbf{f}_t$) and the zero-shot classifier prototypes ($\mathbf{f}_{t, \text{class}}$). The resulting logits from this disentangled text branch are then integrated into the final classification.
>
> This experiment successfully isolates the contribution of ICA-based disentanglement on text domain features), demonstrating that using the recovered independent factors in text consistently boosts few-shot performance, thereby empirically validating the symmetrical recovery of the text latent factors. The results are shown in the following (Left: With FastICA, Right: Without FastICA). These results show the advantage of FASTICA applied to text only.
>
> | Dataset            | 1-shot        | 2-shot        | 4-shot        | 8-shot        | 16-shot       |
> | ------------------ | ------------- | ------------- | ------------- | ------------- | ------------- |
> | **Average**        | 64.62 / 62.40 | 66.51 / 64.66 | 68.28 / 66.56 | 70.10 / 68.47 | 71.81 / 70.36 |
> | **Caltech101**     | 92.45 / 87.10 | 92.90 / 88.40 | 93.43 / 89.17 | 93.55 / 89.66 | 93.75 / 90.14 |
> | **DTD**            | 48.11 / 46.16 | 51.54 / 49.65 | 54.91 / 54.02 | 59.46 / 58.39 | 61.52 / 61.05 |
> | **EuroSAT**        | 60.32 / 55.27 | 64.33 / 61.64 | 69.86 / 65.60 | 72.43 / 68.04 | 75.74 / 70.69 |
> | **FGVC**           | 18.96 / 18.99 | 21.45 / 21.30 | 22.38 / 22.32 | 25.35 / 25.44 | 29.97 / 29.94 |
> | **Food101**        | 78.91 / 77.38 | 78.98 / 77.55 | 79.13 / 77.55 | 79.19 / 77.78 | 79.25 / 77.89 |
> | **ImageNet**       | 60.68 / 60.46 | 60.82 / 60.67 | 60.92 / 60.80 | 61.42 / 61.26 | 61.93 / 61.80 |
> | **Oxford-Flowers** | 73.61 / 73.24 | 79.09 / 78.93 | 84.13 / 83.76 | 88.27 / 88.27 | 89.97 / 89.97 |
> | **Oxford-Pets**    | 85.91 / 86.07 | 87.11 / 86.92 | 86.32 / 86.37 | 87.33 / 87.11 | 88.17 / 88.25 |
> | **Stanford-Cars**  | 61.41 / 57.64 | 62.41 / 58.64 | 64.77 / 61.75 | 66.57 / 63.09 | 68.86 / 66.82 |
> | **SUN397**         | 63.41 / 61.32 | 64.57 / 62.71 | 66.05 / 64.22 | 67.35 / 65.57 | 68.41 / 66.80 |
> | **UCF101**         | 66.98 / 62.78 | 68.41 / 64.87 | 69.18 / 66.61 | 70.24 / 68.54 | 72.30 / 70.63 |

---

> ### Author Response · Authors · 2025-11-24
>
> **Q3: No additional method or component is introduced to correspond to the theory; the paper relies on existing techniques (PCA and FastICA).**
>
> **R3:** We acknowledge the reviewer's observation that our proposed practical methodology utilizes existing techniques, namely PCA and FastICA.
>
> We emphasize that the core contributions of this paper, as highlighted in the Introduction, is not the introduction of a new method, but providing the theoretical foundation, guarantees, and principled guidance for applying these tools to extract disentangled representations from modern MMCL models.
>
> Our contributions are explicitly structured in Introduction as follows
>
> *Model Novelty* We propose a Novel Latent Partial Causal Model to overcome the limitations of traditional DAGs for large-scale multimodal data.
>
> *Theoretical Novelty* We establish an Identifiability Guarantee proving that MMCL representations are related to the latent factors up to simple ambiguities (linear or permutation transformations).
>
> *Disentanglement Potential*: We use this theory to reveal the \textit{Disentanglement Potential of MMCL} for the first time4.Extensive Validation: We provide \textit{Extensive Experimental Results} that validate these theoretical insights on real-world datasets.
>
> The significance of our work lies in the theoretical link established between the complex CLIP-like models and the properties required for these simple linear post-processing tools to be effective. Our consistent empirical success (e.g., performance gains in few-shot learning by incorporating FastICA) demonstrates that this theory-derived pipeline is a novel and effective practical methodology for leveraging the representations of pre-trained models.
>
> **Typos or Mistakes** We thank the reviewer for the careful reading and have corrected the typos accordingly.

---

### Official Review · Reviewer_9bGu · 2025-10-30

**Soundness:** 3
**Presentation:** 3
**Contribution:** 3
**Rating:** 8
**Confidence:** 3

**Summary:**

The authors critique the standard assumption that a single Directed Acyclic Graph (DAG) can model the generative process of large-scale multimodal data. They argue that real-world data (e.g., image-text pairs) often result from heterogeneous processes, including conflicting causal directions like text-to-image and image-to-text. To address this, they propose a novel latent partial causal model. This model separates latent variables into latent coupled variables and modality-specific variables. The paper's core theoretical contribution is an \textbf{identifiability result}: it proves that representations learned by MultiModal Contrastive Learning (MMCL) can recover the true latent coupled variables ($z_x, z_t$) up to a simple transformation.  This theory implies that models like CLIP learn disentangled representations, which are "mixed" by a simple transformation. The authors validate this practically by showing that applying post-hoc FastICA (for the hypersphere case) or PCA + FastICA (for the convex body case) to pre-trained CLIP embeddings improves performance on few-shot learning and domain generalization tasks.

**Strengths:**

1. The paper clearly identifies a valid and important weakness in existing causal models for multimodal learning. The observation that web-scale data is a mixture of generative processes (e.g., text-to-image and image-to-text) is sharp and provides a strong foundation for their new model.

2. The connection between the abstract theory (identifiability of $z_x$) and a simple, practical recommendation (use FastICA on CLIP embeddings) is a significant strength. It makes the theory testable and useful.

3. The experiments are very effective. Showing that a simple, off-the-shelf, post-processing step (FastICA) on a pre-trained CLIP model can improve performance across 16+ real-world datasets is a strong validation of the theory. The qualitative disentanglement on CelebA (Figure 3) is also very illustrative.

**Weaknesses:**

1. The paper's core motivation is replacing DAGs with a model featuring an "undirected edge" between $z_x$ and $z_t$. However, the actual mathematical parameterizations (Eq. 4 \& 7) define a directed conditional probability, $p(z_t|z_x)$. This creates a mismatch between the conceptual claim and the formal execution.

2. The proofs rely on the conclusion that optimal encoders must be \emph{perfectly} independent of modality-specific variables ($m_x, m_t$). This is an extremely strong assumption, derived from a necessary condition for optimality, and is unlikely to hold perfectly in practice.

3. The paper provides two methods (FastICA and PCA + FastICA) based on two different theories but offers no clear heuristic to determine which is appropriate for a given model or dataset.

4.  The paper claims "component-wise" identifiability but validates it with \emph{qualitative} and \emph{semantic} disentanglement (e.g., "Smile"). It lacks standard \emph{quantitative} disentanglement metrics (MIG, SAP, etc.) to support this strong technical claim.

**Questions:**

1. The central modeling innovation is the "undirected edge" between $z_x$ and $z_t$. However, the generative parameterizations in Eq. (4) and Eq. (7) both define a directed conditional probability $p(z_t|z_x)$. How does this parameterization formally represent an "undirected" relationship, as opposed to a standard directed model $z_x \rightarrow z_t$? Is the "undirected" nature meant to imply that the model is a marginalization of a mixture of DAGs (like those in Figure 1), and that $p(z_t|z_x)$ is just one assumed form for this marginalized relationship?

2. In the proofs of Theorem 4.1 and 4.2 (Appendices D.1 and E.1), a critical step relies on Lemma 2. This lemma states that the optimal solution is achieved when $h_x(m_x, z_x) = h_t(m_t, z_t)$ almost surely. By differentiating, the authors conclude that the optimal encoders must be independent of $m_x$ and $m_t$. This implies that the MMCL loss perfectly enforces the encoders to discard all modality-specific information. Is this a reasonable assumption for the proof, or is it an assumed condition for optimality that may not be achievable? How does the identifiability theory hold if there is minor "leakage" of $m_x$ into the representation?

3. The paper offers two practical methods: FastICA for hyperspheres (Corollary 1) and PCA + FastICA for convex bodies (Corollary 2). Practically, how should one choose between these two methods? Is there an empirical test to determine the "geometry" of CLIP's latent space? Corollary 2 (convex bodies) implies the learned representation $f_x(x) = Pz_x + c$ is already disentangled. Why does this "simpler" case require a more complex solution (PCA + FastICA)? The paper states PCA is needed "to account for the orthogonal transformation introduced by PCA," which is circular.


4. The CelebA experiments show compelling semantic disentanglement ("Smile," "Gender"). The theory, however, promises "component-wise" identifiability. Does this result imply that the true, independent latent factors for CelebA \emph{are} these high-level semantic attributes? Have you considered quantitatively measuring disentanglement (e.g., MIG, DCI, SAP) by training a linear classifier on the disentangled representations to predict the ground-truth attributes?

---

> ### Author Response · Authors · 2025-11-24
>
> We sincerely appreciate your time spent reviewing our draft and your positive feedback on our work.
>
> ---
>
> **Q1: Conditional distribution vs. directed edge? a mixture of DAGs?**
>
> **R1:**
> We thank the reviewer for pointing out this critical subtlety, which is indeed central to understanding the nature of our latent partial causal model.
>
> *Statistical Parameterization vs. Causal Semantics.*
>
> It is correct that the parameterizations in Eq. (4) and Eq. (7) use conditional forms such as
> $p(z\_t \mid z\_x)$. However, a conditional distribution is a \emph{statistical representation of
> dependence}, not a specification of causal direction. Even if the true causal direction were
> $a \rightarrow b$, we can still consistently define the reverse conditional $p(a \mid b)$.
> Formally, any joint distribution admits two symmetric factorizations: $p(z_x, z_t)= p(z_x)p(z_t \mid z_x) \qquad \text\{or\} \qquad p(z_t)p(z_x \mid z_t).$ Selecting one decomposition (e.g., $p(z_t \mid z_x)$) is just a mathematical convenience, rather than a claim that the underlying causal graph is directed from $z_x$ to $z_t$.
>
> *How This Relates to the ``Undirected Edge.''*
> The ``undirected'' edge in our latent partial causal model is intended to express symmetric statistical dependence between $z_x$ and $z_t$, without assuming or committing to any causal direction. In this sense, it is conceptually analogous to the dependence encoded in an undirected graphical model (e.g., a Markov random field), where edges represent bidirectional coupling rather than directional causation.
>
> *Relation to Mixtures of DAGs.* We further clarify that the undirected relationship does not assume that the model is literally the marginalization of a mixture of DAGs, although such a mixture may be one intuitive way to illustrate how symmetric dependence can arise. Instead, our model directly specifies an undirected dependency structure at the latent level. The conditional parameterization  $p(z_t \mid z_x)$ used in Eq. (4) and Eq. (7) should therefore be understood as \emph{one valid statistical factorization of the joint}, not as an embodiment of a causal arrow. It does not alter or contradict the fundamentally undirected semantics of the latent coupling.
>
> **Q2: Is optimality a reasonable assumption for the proof, "leakage" of  into the representation?**
>
> **R2:**
> Thank you for the question. We would like to clarify the distinction between identifiability and practical solution attainment:
>
> Identifiability concerns existence: It asks whether a unique solution for the latent variables exists under idealized conditions (e.g., infinite data and perfect optimization). It is fundamentally about the existence of a unique solution, independent of whether it can be reached in practice. If one were to assume that the optimal solution can never be obtained, then identifiability analysis would indeed be meaningless.
>
> Bounds concern attainability: You are correct that, in practice, the optimal solution may not be exactly reachable due to finite data, local optima, or optimization constraints. Bound analyses study how closely one can approximate the ideal solution. This is a separate research line from identifiability and complements it by providing guarantees on the achievability or approximation error.
>
> In other words, identifiability establishes that a solution exists and is unique in principle, while bounds address the practical ability to recover or approximate that solution.
>
> Moreover, our robustness experiments across real CLIP model and datasets, support the notion that our theoretical results remain a strong guiding principle in practice.
>
>
> **Q3: FastICA vs. PCA + FastICA, how should one choose between these two methods?**
>
> **R3:** Thanks for such insightful comments.
>
> Since our theoretical results rely on assumptions about the true generative models, e.g., hyperspheres or convex bodies, and the validity of these assumptions is unknown, it is challenging to provide principled guidance. Nevertheless, in practice, one of the main advantages of learning disentangled representations is to improve downstream task performance. Given this, a practical approach is to treat FastICA and PCA + FastICA as two hyperparameters and select the best option based on validation data. The rationale is that a “good” disentangled representation should enhance performance on related downstream tasks. Empirically, we observed that for face image data, the hypersphere assumption seems reasonable. Specifically, FastICA performs well for learning disentangled representations in our experiments on CelebA, which aligns with prior work showing that mapping face image features improves downstream task performance. More details can be found in Sec.~L.

---

> ### Author Response · Authors · 2025-11-24
>
> **Q5: Is there an empirical test to determine the "geometry" of CLIP's latent space.  Corollary 2 (convex bodies) implies the learned representation is already disentangled. Why does this "simpler" case require a more complex solution (PCA + FastICA)? **
>
> **R5:** We thank the reviewer for this insightful question again.
>
> If we understand correctly, the reviewer is referring to the geometry of the CLIP representation space. CLIP representations are constrained to lie on a hypersphere due to the $L_2$ normalization applied during training.
>
> Corollary 2 states that if (i) the latent variables are generated from convex bodies and (ii) the learned representation also lies in a convex body, then the representation is already disentangled up to an orthogonal transformation. While this guarantees an extremely simple recovery procedure in the convex-body setting, the key issue is that CLIP’s embedding space *violates* assumption (ii): it is not a convex body but a hypersphere. Because of this geometric mismatch, Corollary 2 may not directly apply to CLIP embeddings.
>
> Nevertheless, the insight behind Corollary 2 remains applicable with appropriate adjustments. The corollary relies on the existence of an isometric mapping from the latent convex body to the representation space. While a global isometry between a convex body and a hypersphere is not feasible, it is reasonable to assume a *local* isometry between the convex body and sufficiently small regions of the hypersphere. Motivated by this, we first apply PCA to the spherical representations $\mathbf{f}_x(\mathbf{x})$ to obtain an approximately Euclidean (convex) embedding. This yields a representation that is effectively a convex-to-convex mapping, thereby satisfying assumptions (i) and (ii) of the corollary. However, in this process, PCA may introduce an additional linear transformation. We thus use FastICA to remove the linear transformation introduced by PCA, allowing the recovery of the final disentangled components. This PCA+FastICA pipeline thus enables effective use of CLIP-like representations under the theoretical framework provided by Corollary 2.
>
>
> In short, the case is “simpler’’ from the perspective of identifiability theory (the representation is already disentangled), but it becomes “more complex’’ in practice because CLIP’s hyperspherical geometry violates the assumptions of Corollary~2, motivating the PCA + FastICA correction.
>
>
>
> **Q6: Does this result imply that the true, independent latent factors for CelebA \emph{are} these high-level semantic attributes? Have you considered quantitatively measuring disentanglement (e.g., MIG, DCI, SAP) by training a linear classifier on the disentangled representations to predict the ground-truth attributes?**
>
> **R6:** Since the true latent variables are always unknown, we generally cannot obtain ground truth. Consequently, for any learned representation, we are typically unable to directly quantify how well it matches the true latent variables. Therefore, we may not claim that the learned representations exactly recover the true latent factors.
>
> Nevertheless, as shown in our empirical results on CelebA, the learned representations are interpretable and visually reasonable. We evaluated disentanglement using DCI, obtaining DCI-C: 0.11, DCI-D: 0.06, and DCI-I: 0.82. We note that these metrics can sometimes be controversial [1]. Moreover, we lack directly comparable baselines, since CLIP is trained on large-scale data with a relatively modern architecture compared to existing methods such as TC-VAE, FactorVAE. Given the above, we primarily emphasize qualitative results on CelebA. For quantitative evaluation, we focus on downstream tasks, such as domain generalization and few-shot learning, which provide a practical measure of the utility of the learned representations.
>
> [1] Locatello, Francesco, et al. "A sober look at the unsupervised learning of disentangled representations and their evaluation." Journal of Machine Learning Research 21.209 (2020): 1-62.

---

### Official Review · Reviewer_Mb1E · 2025-10-31

**Soundness:** 2
**Presentation:** 3
**Contribution:** 3
**Rating:** 6
**Confidence:** 1

**Summary:**

The paper proposes a latent partial causal model to generalize beyond traditional DAGs, capturing undirected multimodal dependencies and proving identifiability in contrastive learning frameworks like CLIP.

**Strengths:**

- The extension of the causal structure in multimodal learning beyond DAGs is a valuable conceptual exploration.
- The identifiability proofs are soundly constructed.

**Weaknesses:**

I don't have enough knowledge to evaluate this paper as I don't work in related domains. For me, it seems the paper did not empirically demonstrate that the learned representations encode genuine causal relations between modalities.

**Questions:**

Can the proposed method be tested on text domain datasets?

---

> ### Author Response · Authors · 2025-11-24
>
> We sincerely thank the reviewer for their summary and comments on our paper. We especially appreciate their candidness regarding the domain expertise (Confidence Score 1) and will alert the Area Chair to ensure sufficient expert review of our technical contributions. We address the core points below:
>
> ---
>
> The reviewer points out that the paper does not empirically demonstrate that the learned representations encode genuine causal relations between modalities.
>
> Our core objective is not to prove causal directionality, but identifiability of shared, coupled factors. Our proposed latent partial causal model explicitly moves beyond the strict causal directionality enforced by traditional DAGs. The purpose of introducing the undirected edge is to model the generalized scenario where the latent factors are merely associated or coupled, capturing heterogeneous generative processes (image-to-text, text-to-image) that coexist in large datasets like CLIP's training data. The "causal" aspect of our model lies in the generative process: the latent factors $(\mathbf{z}, \mathbf{m})$ are the causes of the observed data $(\mathbf{x}, \mathbf{t})$. The theoretical contribution is proving that the MMCL framework successfully identifies these latent causes (up to a trivial transformation). Therefore, the success of our method is measured by its ability to extract these disentangled coupled factors, not by determining the specific direction of causality between $\mathbf{z}_x$ and $\mathbf{z}_t$.
>
> ----
>
> Regarding to the Proposed Method be Tested on Text Domain. We would like to clarify that the primary context of this work is the analysis of multimodal learning. Consequently, our main experimental focus has been on multimodal tasks.
>
> Nevertheless, to better demonstrate the disentanglement potential within the text domain, we have included the following additional experiment:
>
>
> Building on Tip-Adapter, we introduce a new text-alignment module. Specifically, for each query image, we generate a caption using the LLaVA model and feed it into the CLIP text encoder. We then apply ICA to the resulting query text feature. The similarity between the ICA-transformed query text feature and the ICA-transformed zero-shot classifier is computed to produce the logits of the text-alignment module. These logits are subsequently added to the original Tip-Adapter logits to obtain the final classification logits. For comparison, we also evaluate Tip-Adapter with the text-alignment module without ICA. This allows us to assess the specific contribution of ICA-based disentanglement on text domain to few-shot learning performance. The results are as follows, Left: With FastICA, Right: Without FastICA.
>
> | Dataset            | 1-shot        | 2-shot        | 4-shot        | 8-shot        | 16-shot       |
> | ------------------ | ------------- | ------------- | ------------- | ------------- | ------------- |
> | **Average**        | 64.62 / 62.40 | 66.51 / 64.66 | 68.28 / 66.56 | 70.10 / 68.47 | 71.81 / 70.36 |
> | **Caltech101**     | 92.45 / 87.10 | 92.90 / 88.40 | 93.43 / 89.17 | 93.55 / 89.66 | 93.75 / 90.14 |
> | **DTD**            | 48.11 / 46.16 | 51.54 / 49.65 | 54.91 / 54.02 | 59.46 / 58.39 | 61.52 / 61.05 |
> | **EuroSAT**        | 60.32 / 55.27 | 64.33 / 61.64 | 69.86 / 65.60 | 72.43 / 68.04 | 75.74 / 70.69 |
> | **FGVC**           | 18.96 / 18.99 | 21.45 / 21.30 | 22.38 / 22.32 | 25.35 / 25.44 | 29.97 / 29.94 |
> | **Food101**        | 78.91 / 77.38 | 78.98 / 77.55 | 79.13 / 77.55 | 79.19 / 77.78 | 79.25 / 77.89 |
> | **ImageNet**       | 60.68 / 60.46 | 60.82 / 60.67 | 60.92 / 60.80 | 61.42 / 61.26 | 61.93 / 61.80 |
> | **Oxford-Flowers** | 73.61 / 73.24 | 79.09 / 78.93 | 84.13 / 83.76 | 88.27 / 88.27 | 89.97 / 89.97 |
> | **Oxford-Pets**    | 85.91 / 86.07 | 87.11 / 86.92 | 86.32 / 86.37 | 87.33 / 87.11 | 88.17 / 88.25 |
> | **Stanford-Cars**  | 61.41 / 57.64 | 62.41 / 58.64 | 64.77 / 61.75 | 66.57 / 63.09 | 68.86 / 66.82 |
> | **SUN397**         | 63.41 / 61.32 | 64.57 / 62.71 | 66.05 / 64.22 | 67.35 / 65.57 | 68.41 / 66.80 |
> | **UCF101**         | 66.98 / 62.78 | 68.41 / 64.87 | 69.18 / 66.61 | 70.24 / 68.54 | 72.30 / 70.63 |

---

### Official Review · Reviewer_QDaK · 2025-11-03

**Soundness:** 3
**Presentation:** 2
**Contribution:** 3
**Rating:** 6
**Confidence:** 3

**Summary:**

The paper proposes a Latent Partial Causal Model for multimodal data (image and text). Within this framework, the authors establish identifiability guarantees between the learned multimodal contrastive representations and the underlying generative latent factors. Specifically:

- When the true generative factors lie on a unit hypersphere, the learned representations are identifiable up to a linear transformation.

- When the true generative factors lie in a convex body, the learned representations are identifiable up to permutation, scaling, and shifting of the true generative factors.

These results imply that multimodal contrastive learning (MMCL), such as CLIP, has the potential to learn disentangled and meaningful representations rather than arbitrary embeddings.
In experiments, the authors first validate these theoretical claims through synthetic experiments. They then apply their method to pretrained CLIP embeddings and demonstrate that the resulting representations improve performance across various downstream tasks.

**Strengths:**

**S1.** The proposed model is interesting and suitable for multimodal settings.

**S2.** The identifiability analysis is solid, and the techniques used in the proofs are interesting.

**S3.** The downstream experiments using disentangled representations span several application areas, and the learned representations demonstrate better qualities compared to the baseline.

**Weaknesses:**

**W1.** The gap between the theoretical analysis and practical use is not clear. **The identifiability results rely on several assumptions** in Equations (4) and (7) (especially the probability density assumptions), along with assumptions “the latent space is the unit hypersphere” / “the latent space is a hyperrectangle.” **However, it is unclear whether the assumptions hold.** If I understand correctly, these assumptions are made on the ground-truth underlying generative factors. I suggest adding more discussion of what these assumptions mean at a high level (not just mathematically), why they are reasonable, and in which situations they may be violated.

**W2**. There is a lack of a formal definition of disentanglement. The paper highlights that a key practical use of the theory is to obtain disentangled representations using a pretrained multimodal encoder (e.g., CLIP). However, the discussion in Section 5 is high-level and lacks mathematical grounding, which could lead to misunderstandings. For example, the intended disentanglement is among the components of $z_x$ (or $z_t$), not disentanglement between $z$ and the modality-specific noise $m$, correct? I suggest including a precise definition of disentanglement in the paper and discussing it more clearly.

**W3.** The validation of the identifiability results is based on low-dimensional synthetic experiments, and the evaluation of disentangled representations on real-world data is done purely through downstream tasks. I suggest adding a semi-synthetic dataset (e.g., Causal3DIdent[1], Morph-MNIST[2]) that contains higher-dimensional ground-truth latent variables, so disentanglement can be evaluated directly rather than indirectly via downstream performance.

[1] Von Kügelgen, Julius, et al. "Self-supervised learning with data augmentations provably isolates content from style." Advances in neural information processing systems 34 (2021): 16451-16467.

[2] Castro, Daniel C., et al. "Morpho-MNIST: Quantitative assessment and diagnostics for representation learning." Journal of Machine Learning Research 20.178 (2019): 1-29.

**Questions:**

**Q1** Can the authors explain more about the aggregated functions h_x and h_t, and how they are used in the proof (Lines 964–977)? Is h_x(m_x, z_x) or h_x(z_x) actually h_x(m_x, z_x, m_t, z_t)? Also, are there smoothness or differentiability assumptions on h required for the proof?

**Q2**. In Corollaries 1 and 2, the learned representations are an invertible linear transformation of the ground-truth vectors, or a permutation after scaling and shifting. **Does this mean the dimensionality of the learned representations must match the dimensionality of the ground-truth latent variables?** If so, this seems impractical in real-world settings (e.g., using a pretrained encoder like CLIP, where we do not know the latent dimensionality). **Can the theorem be generalized when there is a dimensionality mismatch between the learned representations and the ground truth? Does this mismatch affect the practical applicability of the method?**

**Q3.** It appears that the proposed partial causal model (Figure 2) subsumes all the DAG structures shown in Figure 1 and can represent any setting where there is some correlation between $z_x$ and $z_t$. If so, why is this model considered “causal,” and why is it described as partial causal rather than a general latent-variable model?

---

> ### Author Response · Authors · 2025-11-24
>
> Before responding, we would like to sincerely thank you for taking the time to review our work. In particular, your constructive suggestions have been extremely helpful in improving our manuscript.
>
>
> ----
>
> **Q1: Discussion Regarding to the assumptions used**
>
> **R1:**
> We appreciate the reviewer's insightful comment regarding the gap between theoretical analysis and practical use, which indeed represents a fundamental challenge in the field of identifiability analysis
>
> We fully agree that the true underlying data generating process (DGP) is fundamentally unknown in real-world scenarios. It is a common challenge in latent variable modeling that, without imposing certain assumptions, the identifiability becomes impossible, even for relatively simple cases like nonlinear independent component analysis. Consequently, almost all related work concerning the identifiability of latent variables have to enforce specific, well-defined assumptions to proceed with rigorous theoretical analysis. As rightly noted, verifying these specific assumptions (e.g., regarding the exact distributions or the geometry of the latent space in Equations (4) and (7)) against real-world DGPs is a major, common challenge facing the entire research community, given the inherent unknown nature of DGP.
>
> In the revised version, we have added a section in the Appendix (Sec. N) that discusses the assumptions from a high-level perspective, outlines their limitations, and describes potential practical application scenarios.
>
> **Q2: A precise definition of disentanglement**
>
> **R1:** We thank the reviewer for pointing out the need for a more formal definition of {disentanglement}. The term "disentanglement" in the context of our paper refers to the standard definition used in Independent Component Analysis (ICA) and Causal Representation Learning: the ability to recover the individual, mutually independent components of the coupled factors.
>
>
> Mathematical Grounding: The mathematical descriptions provided in \textbf{Corollary 1} ($\mathbf{f}\_x(\mathbf{x}) = \mathbf{A} \mathbf{z}\_x + \mathbf{c}$, with $\mathbf{A}$ being an orthogonal matrix) and Corollary 2 ($\mathbf{f}\_x(\mathbf{x}) = \mathbf{P} \mathbf{z}\_x + \mathbf{c}$, with $\mathbf{P}$ being a permutation matrix with scaling) define the relationship between the learned representation $\mathbf{f}\_x(\mathbf{x})$ and the true latent variable $\mathbf{z}\_x$ \textbf{up to a simple transformation (linear or permutation).
>
> We agree that formalizing this concept is vital. In the revised version of the manuscript, we have included a formal definition of Disentanglement in Appendix O. Thanks a lot.
>
> **Q3: Semi-synthetic imagedataset**
>
> **R3:** We thank the reviewer for this insightful suggestion. Considering multimodal settings, we conducted experiments on the *Multimodal3DIdent* dataset, which provides higher-dimensional data with complete ground-truth latent factors, see Sec. P for details in the new version. This allows us to directly evaluate the identifiability of the latent variables ($\mathbf{z}_x$ and $\mathbf{z}_t$) across image and text modalities. As shown in the Table in Sec. P, our image branch ($\mathbf{z}_x$) achieves good recovery ($R^2 \approx 0.97$) of the shared latent factors, while the text branch ($\mathbf{z}_t$) achieves slightly lower performance ($R^2 \approx 0.75$), likely due to the discrete nature of text factors, which violates the continuous assumption in our theoretical framework. Overall, these results provide direct empirical support that our linear identifiability results extend to higher-dimensional image data. We thank the reviewer for this suggestion, which substantially strengthens our work.

---

> ### Author Response · Authors · 2025-11-24
>
> **Q4: Explain more about the aggregated functions $\mathbf{h}\_x$ and $\mathbf{h}\_t$**
>
> **R4:** We appreciate the reviewer's detailed inquiry into the core mechanism of our proof, which allows us to clarify the mathematical steps underpinning the separation of variables.
>
> 1. Definition and Input of $\mathbf{h}_x$ and $\mathbf{h}_t$The functions $\mathbf{h}_x$ and $\mathbf{h}_t$ are defined as the Aggregated Functions for their respective modalities, representing the entire transformation pipeline from the potential latent factors to the final learned representation:
>
> Image Modality ($\mathbf{x}$):$$\mathbf{h}_x \stackrel{\text{def}}{=} \mathbf{f}_x \circ \mathbf{g}_x$$Here, $\mathbf{g}_x$ is the image generative process that takes the coupled variable $\mathbf{z}_x$ and the modality-specific variable $\mathbf{m}_x$ as input (i.e., $\mathbf{x} = \mathbf{g}_x(\mathbf{z}_x, \mathbf{m}_x)$)1. $\mathbf{f}_x$ is the learned image encoder. Thus, $\mathbf{h}_x$ takes the inputs that generate $\mathbf{x}$: $\mathbf{h}_x(\mathbf{z}_x, \mathbf{m}_x)$.
>
> Text Modality ($\mathbf{t}$):$$\mathbf{h}_t \stackrel{\text{def}}{=} \mathbf{f}_t \circ \mathbf{g}_t$$. Similarly, $\mathbf{h}_t$ takes its potential inputs: $\mathbf{h}_t(\mathbf{z}_t, \mathbf{m}_t)$.
>
> In summary: $\mathbf{h}_x$ only depends on $(\mathbf{z}_x, \mathbf{m}_x)$, and $\mathbf{h}_t$ only depends on $(\mathbf{z}_t, \mathbf{m}_t)$.
>
>
> Given the above, in Lines 964–977, the Step I utilizes the assumed independence between the coupled ($\mathbf{z}$) and modality-specific ($\mathbf{m}$) latent factors in our generative model to prove that MMCL inherently filters out $\mathbf{m}$. Specifically, we start with the necessity condition, derived from minimizing the contrastive loss $\mathbf{h}_x(\mathbf{m}_x, \mathbf{z}_x) = \mathbf{h}_t(\mathbf{m}_t, \mathbf{z}_t) \quad \text{almost surely}$. We then apply partial differentiation with respect to the noise factor $\mathbf{m}_x$ on both sides:$\frac{\partial \mathbf{h}_x(\mathbf{m}_x, \mathbf{z}_x)}{\partial \mathbf{m}_x} = \frac{\partial \mathbf{h}_t(\mathbf{m}_t, \mathbf{z}_t)}{\partial \mathbf{m}_x}$. Since $\mathbf{m}_x$ is independent of $\mathbf{h}_t$ and its inputs $(\mathbf{m}_t, \mathbf{z}_t)$ in the proposed generative model, the right-hand side must be $\frac{\partial \mathbf{h}_t(\mathbf{m}_t, \mathbf{z}_t)}{\partial \mathbf{m}_x}=\mathbf{0}$. This forces the left-hand side to $\frac{\partial \mathbf{h}_x(\mathbf{m}_x, \mathbf{z}_x)}{\partial \mathbf{m}_x} =\mathbf{0}$: $\frac{\partial \mathbf{h}_x}{\partial \mathbf{m}_x} = \mathbf{0}$. A zero partial derivative implies that $\mathbf{h}_x$ must be independent of its modality-specific input $\mathbf{m}_x$, leading to: $\mathbf{h}_x(\mathbf{m}_x, \mathbf{z}_x) = \mathbf{h}_x(\mathbf{z}_x)$. A symmetric argument proves the same for the text modality: $\mathbf{h}_t(\mathbf{m}_t, \mathbf{z}_t) = \mathbf{h}_t(\mathbf{z}_t)$.
>
> This step shows that MMCL achieves aligned cross-modal representations by requiring the factors ($\mathbf{m}_x, \mathbf{m}_t$) to be removed to in order to minimize the contrastive loss. It reduces the problem to identifying the transformation $\mathbf{h}$ that only operates on the shared coupled variables $\mathbf{z}_x$ and $\mathbf{z}_t$.
>
>
> Indeed, the proof relies on differentiability assumptions on the aggregated functions $\mathbf{h}$ to eliminate the modality-specific nuisance variables. We have modified the related part to higlight this, e.g., the mapping from latent to observed to be differentiable. Thanks again.
>
>
> **Q5: Dimensionality match v.s. mismatch**
>
> Thank you for this insightful comment. Similar to most works on identifiability analysis, our theoretical results currently rely on the assumption that the representation dimensionality matches that of the latent variables. This condition ensures that exact recovery of the latent factors is theoretically possible. To the best of our knowledge, this constraint is fundamental to most identifiability theorems in the field of identifiability analysis. Only very few works provide identifiability guarantees under the weaker condition where the representation dimension exceeds the true latent dimensionality; one such example is the work of Sorrenson et al. (2020) (Sorrenson, Peter, Carsten Rother, and Ullrich Köthe. "Disentanglement by nonlinear ica with general incompressible-flow networks (gin)."). It is interesting to explore generalizing our current results, but is nontrivial.
>
> Moreover, even when there is a dimensionality mismatch, our results appear to remain plausible. This is supported by our empirical observations. In practice, the representation dimension of pre-trained CLIP rarely matches the true latent dimensionality. Nevertheless, across our experiments, the theoretical insights appear to remain valid: applying FastICA or PCA+FastICA provides disentanglement results and yields performance gains across diverse downstream tasks and datasets.

---

> ### Author Response · Authors · 2025-11-24
>
> **Q6: why is this model considered “causal,” and why is it described as partial causal rather than a general latent-variable model?**
>
> Our motivation stems from the fact that enforcing a fully causal model with strict DAG constraints (i.e., directed edges) may be often restrictive for accurately capturing the heterogeneous generative processes of large-scale multimodal data. To relax this constraint, we introduce the undirected edge between the shared latent factors. This structural feature, which represents mere association or coupling rather than strict directionality, is the key reason we refer to our formulation as a \textbf{partial causal model}. In the causality literature, such partial causal structures are commonly represented using Partial DAGs, where some edges may be undirected, unspecified, or uncertain.

---

### Meta-Review · Area_Chair_6zmX · 2025-12-30

**Summary:**

This paper considers learning disentangled representation in multi-modal setting via contrastive learning. The major contribution is to consider two-modality CL problem via a class of DAGs defined across the modalities, and show how an approximated form of contrastive learning objective (CLIP-like) can identify the latent causes (correlated between modalities) under certain generative model assumptions.

Reviewers' major concerns include unclear bits of the theoretical results as well as experimental settings (see below box). Many of them are addressed, some are outstanding still. However the contributions are clear and the paper would be an interesting add to the conference proceedings, targeting audience in the field of causal representation learning.

In the camera ready, I suggest the authors to change the title by removing "beyond DAGs": your method is still based on a DAG but this DAG is defined on multiple modalities, rather than using different and disconnected DAGs for each modality. This is also suggested by Reviewer 9bGu.

**Reviewer Concerns:**

Concerns addressed:
- Lack of formal definition of disentanglement. In rebuttal the authors described it based on the DAG assumption as well as the identifiability of the $z$ variables for different modalities. The AC views it as the authors' definition of disentanglement and considers this concern as being addressed, although the AC also notes that a good definition of disentanglement is still an open and debatable research question.
- Whether empirical results on disentanglement can also be demonstrated on text modality. In rebuttal the authors conducted one experiment to address this issue.

Concerns outstanding:
- Unclear about whether and to what extent the theoretical assumptions are practical, and how the designed algorithm and network architectures align with the assumptions.
- Experimental results lack quantitative metrics used in disentangled representation learning literature (e.g., MIG).

**Reviewer Scores:**

Apart from Reviewer Mb1E who effectively didn't understand the paper, I think other reviewers' initial scores are high enough, so likely they wouldn't have change the score after the rebuttal.

---

### Decision · Program_Chairs · 2026-01-26

Accept (Poster)